# MDL-Pool: Adaptive Multilevel Graph Pooling Based on Minimum Description Length

## Abstract

Graph pooling compresses graphs and summarises their topological properties and features in a vectorial representation. It is an essential part of deep graph representation learning for graph-level tasks like classification or regression. Current approaches pool hierarchical structures in graphs by iteratively applying shallow pooling operators up to a fixed depth. However, they disregard the interdependencies between structures at different hierarchical levels and do not adapt to datasets that contain graphs with different sizes that may require pooling with various depths. To address these issues, we propose MDL-Pool, a pooling operator based on the minimum description length (MDL) principle, whose loss formulation explicitly models the interdependencies between different hierarchical levels and facilitates a direct comparison between multiple pooling alternatives with different depths. MDL-Pool builds on the map equation, an information-theoretic objective function for community detection, which naturally implements Occam's razor and balances between model complexity and goodness-of-fit via the MDL. We demonstrate MDL-Pool's competitive performance in an empirical evaluation against various baselines across standard graph classification datasets.

## 1 Introduction

Graph neural networks (GNNs) have been applied to graph-structured data from various domains to address diverse research questions, including analysing scientific collaborations (Yanardag & Vishwanathan, 2015) or understanding molecule properties such as mutagenicity (Borgwardt et al., 2005; Debnath et al., 1991). Important applications in graph representation learning are graph-level tasks like classification or regression, which require coarsening a graph to an embedding vector that captures its relevant properties. Learning pooling operators for graph classification usually involves a two-part loss function: An unsupervised part to quantify the goodness of the produced graph clusters, and a supervised part to measure the classification performance.

The creation of links in real-world graphs can often be modelled as a stochastic process (Newman, 2001; Vázquez, 2003; Smiljanić et al., 2023) that is typically not directly observable, connecting nodes in clusters, also called communities (Newman, 2012; Fortunato, 2010). Such communities can be groups of friends in social networks or functional groups amongst molecules (Girvan & Newman, 2002; Borgwardt et al., 2005). Network scientists have developed many methods to analyse the community structure of graphs, differing in how precisely they characterise what constitutes a community (Karrer & Newman, 2011; Peixoto, 2019; Blondel et al., 2008; Traag et al., 2019; Rosvall et al., 2009; Peixoto, 2023; Peixoto & Kirkley, 2023; Fortunato, 2010). Recently, several deep-learning-based approaches have adapted such characterisations of communities, learning clusters in an end-to-end fashion and using them to coarsen graphs for downstream tasks (Bianchi et al., 2020; Ying et al., 2018; Blöcker et al., 2024; Castellana & Bianchi, 2025).

Ying et al. (2018) suggest coarsening graphs by iteratively applying shallow pooling to capture the multilevel clusters found in many real-world graphs (Lancichinetti et al., 2009; Schaub et al., 2023). While this *stacking* of shallow pooling enables end-to-end learning of multilevel clusters, it neglects the interdependencies between different levels when losses at different levels are merely summed up: optimisation via gradient descent considers each level separately but cannot reflect interdependencies between levels. Moreover, this approach fixes the depth, making it unsuitable for empirical graph datasets containing graphs of various depths.

To address these gaps, we propose *MDL-Pool*, an adaptive hierarchical and entropy-based pooling operator that jointly optimises the clusters across different hierarchical levels via explicitly modelling their interdependencies in an integrated loss formulation. Following the minimum description length (MDL) principle, it automatically selects the optimal pooling depth for each graph instance in the dataset. *MDL-Pool* builds on the multilevel map equation, an information-theoretic objective function for community detection (Rosvall & Bergstrom, 2011) that has recently been integrated with GNNs (Blöcker et al., 2024). Our contributions can be summarised as follows:

1. We adapt the map equation for pooling and derive a multilevel loss to learn hierarchical pooling operators jointly optimised across clustering levels. Different from previous works where only coarser levels depend on finer ones, we jointly consider all levels.
2. Our approach implements Occam's razor, selects the optimal number of clusters automatically, and balances between model complexity and fit—all via following the MDL principle. Explicit regularisation, while essential for other approaches, is superfluous in our case.
3. Thanks to the MDL principle, pooling operators with different depths are directly comparable, allowing us to consider various depths in parallel and dynamically select the optimal depth for each graph instance.
4. We empirically verify the utility of *MDL-Pool*'s integrated loss formulation by comparing it to various baseline methods across standard graph pooling benchmarks.

To the best of our knowledge, our work is the first to consider adaptive multilevel graph pooling based on an integrated notion of hierarchical clusters and the first application of the information-theoretic map equation as a pooling operator.

## 2 RELATED WORK

**Minimum description length.**    The minimum description length principle (MDL) is an information-theoretic and compression-based tool for model selection. It states that the best model for a dataset $D$ is the model that minimises the overall description of (i) the model itself and (ii) the data, given the model (Rissanen, 1978; Grünwald et al., 2005). Formally, the MDL selects the model $M^* = \arg\min_M \mathcal{L}(M) + \mathcal{L}(D \mid M)$, where $\mathcal{L}(M)$ is the model's description length and $\mathcal{L}(D \mid M)$ is the description length of the data, given model $M$. $\mathcal{L}(M)$ can be interpreted as the complexity of the model and $\mathcal{L}(D \mid M)$ as how well the model explains the data. Effectively, the MDL implements Occam's razor and facilitates balancing between model complexity and goodness of fit.

**Graph clustering:**    Graph clustering, known as community detection in network science, aims to partition graphs into clusters of "similar" nodes, also called communities (Fortunato, 2010). However, there are many definitions of what "similar" means. Most works adopt the notion that link densities within clusters should be higher than between clusters, but specifics differ: The so-called modularity criterion compares the intra-cluster link density against that in a randomised version of the network (Newman, 2006). Modularity maximisation detects communities by maximising the modularity measure (Blondel et al., 2008; Traag et al., 2019). The stochastic block model (SBM), originally a generative model, assumes that the connectivity between nodes is determined purely by the nodes' cluster memberships, where intra-cluster and inter-cluster links exist pairwise independently with probability $p$ and $q$, respectively (Karrer & Newman, 2011; Peixoto, 2018). Via Bayes' rule, the SBM becomes an inferential approach that detects communities by finding model parameters that maximise the observed links' likelihood. The map equation is based on the MDL and detects communities by searching for patterns in the statistical properties of the stationary distribution of a random walk, exploiting the information-theoretic duality between compression and data regularities (Rosvall et al., 2009; Smiljanić et al., 2023). Spectral and cut-based graph clustering methods (Goldschmidt & Hochbaum, 1994; Shi & Malik, 2000) generalise the 2-cut problem where, given two nodes $u$, $v$, the task is to find a minimal-weight cut to partition the graph into two disconnected parts, one containing $u$ and the other $v$.

Several of these methods were adapted for optimisation with GNNs through gradient descent: Bianchi et al. (2020) learn clusters by optimising a min-cut objective, Tsitsulin et al. (2023) follow the modularity objective, and Blöcker et al. (2024) adapted the map equation. Ying et al. (2018) defined DiffPool, a heuristic that seeks to group nearby nodes while each node should belong to a single cluster. Embedding-based approaches (Grover & Leskovec, 2016; Perozzi et al., 2014) detect communities by first learning a node embedding, followed by k-means clustering in the embedding space.

**Graph pooling:**  Graph pooling creates coarse-grained representations of graphs. Mean and sum pooling coarsen graphs in a single step by taking the mean or sum over the features of all nodes, respectively; however, their simplicity ignores the structure encoded in the graph's links. Score- or one-every-$k$-based techniques select the information of important nodes (Lee et al., 2019; Gao & Ji, 2019). Clustering-based methods detect clusters in the graph and aggregate nodes that belong to the same cluster into super-nodes for coarser representations (Ying et al., 2018; Castellana & Bianchi, 2025). Any shallow pooling approach can be stacked for hierarchical clustering by iteratively applying it until the graph is sufficiently coarse (Ying et al., 2018). However, merely stacking pooling operators does not consider the interdependencies between clusters at different levels because the clustering objective is optimised at finer levels before considering coarser levels. Hence, coarser clusters cannot influence finer ones, meaning that they generally do not capture the characteristics of the latent hierarchical data generation process. Here, we remedy this issue by proposing a hierarchical pooling approach that jointly optimises the clusters across multiple levels.

## 3  BACKGROUND

**Hierarchical graph pooling.**  Grattarola et al. (2024) cast graph pooling into a general framework, involving three operators: select, reduce, and connect. A pooling operator $\text{POOL} : (\mathbf{A}, \mathbf{X}) \to (\mathbf{A}', \mathbf{X}')$ maps the adjacency matrix $\mathbf{A}$ and node features $\mathbf{X}$ to a coarsened adjacency matrix $\mathbf{A}'$ and coarsened node features $\mathbf{X}'$. The three sub-operations are defined as follows:

**Select (SEL)** creates a new reduced set of nodes, called supernodes, and maps the original nodes to these supernodes. *Score-based* methods rank the nodes and keep a certain fraction that become the supernodes, *clustering-based* methods group nodes into clusters and aggregate them into supernodes.

**Reduce (RED)** shrinks the feature matrix either by aggregating the features of the same cluster or by selecting the subset that belongs to the highest ranked nodes.

**Connect (CON)** creates the new graph structure by connecting the supernodes. This operation is guided by the selection matrix and the original topology.

As suggested by Ying et al. (2018), this framework can be applied $n$ times to achieve hierarchical pooling: $\text{POOL}^{(n)} = \text{POOL} \circ \text{POOL}^{(n-1)}$. We refer to this approach as *stacking-based* hierarchical pooling. Notably, the interdependencies between different levels are not considered in this approach because each pooling operator is applied independently. In contrast, we propose a *jointly optimised* hierarchical pooling operator that considers the interdependencies between different levels.

**The Map Equation.**  The map equation is an information-theoretic objective function for community detection based on the MDL principle (Rosvall et al., 2009; Rosvall & Bergstrom, 2011; Smiljanić et al., 2023). It builds on the idea that identifying regularities in data enables efficient compression of that same data. Using the statistics of a random walk at ergodicity as a proxy for the graph's structure, the map equation framework searches for a partition of the nodes into communities, also called *modules*, that enables the most efficient compression of random walks in expectation.

Let $G = (V, E)$ be a graph with nodes $V$ and links $E$. Without modules, the minimum expected cost in bits for describing a random walker step is the entropy over the nodes' ergodic visit rates, $\mathcal{L}_0 = \mathcal{H}(P) = \sum_{u \in V} p_u \log_2 p_u$ (Shannon, 1948). Here, $\mathcal{H}$ is the Shannon entropy, $P = \{p_u \mid u \in V\}$ is the set of node visit rates, and $p_u$ is node $u$'s visit rate. The node visit rates can be computed in closed form for undirected graphs, or with PageRank (Gleich, 2015) or smart teleportation (Lambiotte & Rosvall, 2012) for directed graphs (see Appendix A for details and an example).

With nodes grouped into modules, the expected number of bits to describe a random walker step—also called the *codelength*—becomes a weighted average of the modules' entropies and the entropy at the so-called index level for transitions between modules. The standard map equation (left) computes the codelength for non-hierarchical partitions; the multilevel map equation (right) generalises to hierarchical partitions with $\ell$ levels via recursion (Rosvall & Bergstrom, 2011; Smiljanić et al., 2023):

$$\mathcal{L}_1(\mathsf{M}) = q\mathcal{H}(Q) + \sum_{\mathsf{m} \in \mathsf{M}} p_{\mathsf{m}} \mathcal{H}(P_{\mathsf{m}}) \qquad \mathcal{L}_\ell(\mathsf{M}) = q\mathcal{H}(Q) + \sum_{\mathsf{m} \in \mathsf{M}} \mathcal{L}_{\ell-1}(\mathsf{m}). \qquad (1)$$

Here, $\mathsf{M}$ is the set of modules, $q = \sum_{\mathsf{m} \in \mathsf{M}} q_{\mathsf{m}}$ is the overall module entry rate, $q_{\mathsf{m}}$ is module $\mathsf{m}$'s entry rate; $p_{\mathsf{m}} = \mathsf{m}_{\text{exit}} + \sum_{u \in \mathsf{m}} p_u$ is the fraction of time the random walker spends in module $\mathsf{m}$, and $\mathsf{m}_{\text{exit}}$ is module $\mathsf{m}$'s exit rate. $Q = \{q_{\mathsf{m}}/q \mid \mathsf{m} \in \mathsf{M}\}$ is the set of normalised module entry rates,

and $P_{\mathsf{m}} = \{\mathsf{m}_{\text{exit}}/p_{\mathsf{m}}\} \cup \{p_u/p_{\mathsf{m}} \mid u \in \mathsf{m}\}$ is the set of normalised node visit and exit rates for $\mathsf{m}$. We provide formal definitions of these quantities in Appendix A. Detecting communities with the map equation is done by searching over the possible partitions of nodes into modules to minimise Equation (1), which is a classical NP-hard optimisation problem (Edler et al., 2017; Smiljanić et al., 2023). Note that, for clarity and to reflect the number of pooling steps, we adopt a zero-based naming convention for the map equation, whereas the map equation literature typically uses a one-based naming convention, referring to the case without communities as "one-level", the non-hierarchical case "two-level", and so on.

Why does the map equation work? Via the MDL, two competing objectives interact: First, small modules are desirable as they have lower entropy, enabling cheap descriptions of intra-module steps; however, creating small modules results in many modules. Second, fewer modules are desirable because this leads to fewer module changes, which are expensive to describe; however, using fewer modules means creating large modules with higher entropy and expensive intra-module steps. In practice, these two competing objectives implement Occam's razor, and a tradeoff is required to balance them, automatically selecting the optimal number of modules and levels.

## 4    ADAPTIVE MULTILEVEL POOLING WITH MAP EQUATION LOSS

Depending on their specific approach, pooling operators satisfy different properties. For example, most operators require extensive hyperparameter tuning while parameter-free approaches learn their settings, such as the optimal number of clusters or levels, in a data-driven fashion. Before developing our method, we first identify desirable properties of pooling operators and use them to characterise the pooling operators used in this work in Table 1.

**Learnable**  Learnable operators are essential because pooling is an optimisation problem depending on features and the downstream task. Recent works on pooling heavily utilise learnable operators (Ying et al., 2018; Gao & Ji, 2019; Diehl, 2019; Bacciu et al., 2023; Bianchi et al., 2020; Tsitsulin et al., 2023; Bianchi, 2023; Castellana & Bianchi, 2025).

**Interpretable**  Most methods learn pooling operators in a supervised or self-supervised fashion via downstream tasks or a reconstruction loss. In contrast, we propose learning pooling operators in an unsupervised fashion with a loss function that can be interpreted as the minimum description length of both community structures and graph topology.

**Parameter-free**  The true number of communities and hierarchical levels in empirical graphs is generally unknown and infeasible to obtain (Peel et al., 2017). In practice, the number of communities is chosen based on prior knowledge or via hyperparameter tuning, which is typically computationally expensive. In contrast, parameter-free approaches learn the optimal number of clusters from the data, in our case via the MDL principle acting as a model-selection criterion.

**Hierarchical**  Many empirical graphs have a hierarchical structure (Lancichinetti et al., 2009; Schaub et al., 2023), which previous works capture by stacking pooling operators (Ying et al., 2018; Gao & Ji, 2019). However, they merely sum the losses across multiple levels without considering the interdependencies between these levels. Different from previous approaches, we explicitly model the interdependencies between hierarchical levels in our loss.

**Adaptive Depth**  Stacking-based pooling methods require choosing a specific depth, which acts as a hyperparameter. Our method learns pooling operators for all depths up to $\ell$ and dynamically selects the appropriate depth for each graph using the MDL principle (see Figure 1).

Table 1: Properties of pooling operators used in the empirical evaluation.

| Method | Learnable | Interpretable | Parameter-free | Hierarchical | Adaptive Depth |
|---|---|---|---|---|---|
| Graclus | | | ✓ | | |
| Top-$k$, ECPool, k-MIS | ✓ | | | | |
| MinCut, DMoN, JBGNN | ✓ | ✓ | | | |
| DiffPool | ✓ | ✓ | | (✓) | |
| BNPool | ✓ | ✓ | ✓ | | |
| MDL-Pool (ours) | ✓ | ✓ | ✓ | ✓ | ✓ |

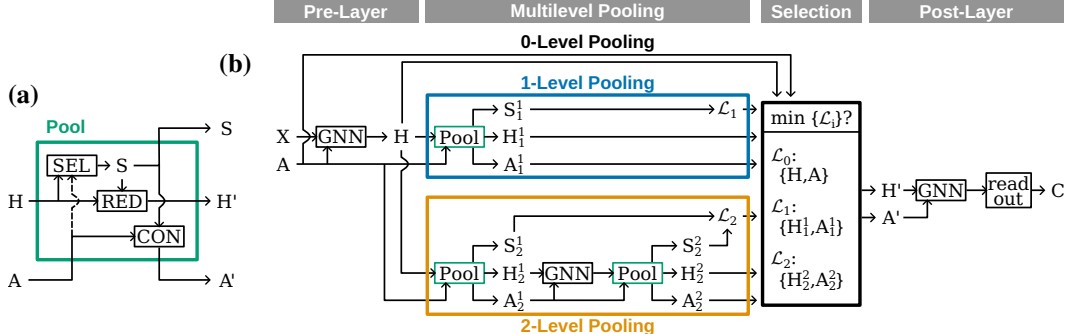

Figure 1: **(a)** Generic pooling block based on SEL-RED-CON. **(b)** Our multilevel pooling setup, here with up to two levels. Matrix subscripts and superscripts denote the pooling depth and number of performed pooling steps, respectively. For example, $\mathbf{S}_2^1$ is the cluster assignment matrix in the 2-level pooling case after 1 pooling step. $\mathcal{L}_0 = \mathcal{H}(P)$ is the no-pooling codelength. An extension to more levels is possible due to our adaptable loss function (see Appendix A.2).

## 4.1 POOLING ARCHITECTURE

**Pooling Building Block:** Like other clustering-based pooling methods, our approach can be cast in the Select-Reduce-Connect framework (Grattarola et al., 2024) whose generic setup is shown in Figure 1(a). The SEL step can generally be implemented via an MLP or a GNN and creates a soft cluster assignment matrix $\mathbf{S} \in \mathbb{R}^{n \times c}$ for $n$ nodes and at most $c$ clusters with learnable parameters $\Theta$ and $\mathrm{softmax}$ activation. In our work, we utilise an MLP, so SEL actually does not depend on the adjacency matrix $\mathbf{A}$. RED and CON use $\mathbf{S}$ to coarsen the node embeddings $\mathbf{H}$ and the adjacency matrix $\mathbf{A}$, respectively,

$$\text{SEL}\left(\mathbf{A}, \mathbf{H}\right) := \mathrm{softmax}\left(\text{MLP}_\Theta(\mathbf{H})\right), \quad \text{RED}\left(\mathbf{S}, \mathbf{H}\right) := \mathbf{S}^T \mathbf{H}, \quad \text{CON}\left(\mathbf{S}, \mathbf{A}\right) := \mathbf{S}^T \mathbf{A} \mathbf{S}. \quad (2)$$

The learnable parameters $\Theta$ define the characteristics of the pooling operation, creating soft assignments $\mathbf{S}$, pooled embeddings $\mathbf{H}'$, and a pooled adjacency matrix $\mathbf{A}'$,

$$\text{POOL}\left(\mathbf{A}, \mathbf{H}\right) := \left\{\text{SEL}\left(\mathbf{A}, \mathbf{H}\right), \text{CON}\left(\text{SEL}\left(\mathbf{A}, \mathbf{H}\right)\right), \text{RED}\left(\text{SEL}\left(\mathbf{A}, \mathbf{H}\right), \mathbf{A}\right)\right\}. \quad (3)$$

To learn the pooling operator, we use a continuous generalisation of the multilevel map equation as an unsupervised training loss, which we derive in Section 4.2. Our loss $\text{L}_\Theta$ follows the minimum description length principle to select the desired pooling operator with $\Theta_{\text{MDL}} = \arg\min_\Theta(\text{L}_\Theta)$.

**Hierarchical pooling:** Learning hierarchical pooling operators involves two parts. First, devising an architecture that stacks pooling building blocks. And second, a hierarchical loss formulation. For hierarchical pooling with $\ell$ levels, we stack pooling layers followed by GNN layers. For $l = 1$ we apply the normal pooling operator once, while for $l = 2, \ldots, \ell$, pooling block $l$ depends on the output of pooling block $l - 1$ via the pooled features and adjacency.

$$\text{POOL}_1^{(1)}(\mathbf{A}, \mathbf{H}) := \text{POOL}(\mathbf{A}, \mathbf{H}), \quad \text{POOL}_\ell^{(l)}(\mathbf{A}, \mathbf{H}) := \text{POOL}(\text{GNN}(\text{POOL}_\ell^{(l-1)}(\mathbf{A}, \mathbf{H}))). \quad (4)$$

To adapt shallow pooling losses for hierarchical pooling, Ying et al. (2018) suggest using the same loss at every pooling step $1 \ldots \ell$ for learnable parameters $\Theta_1 \ldots \Theta_\ell$ and summing the losses, that is,

$$\text{L}_\Theta^{(\ell)} := \text{L}_{\Theta_1} + \text{L}_{\Theta_{1,2}} + \cdots + \text{L}_{\Theta_{1\ldots\ell}} \quad (5)$$

with $\Theta_{1\ldots\ell} := \{\Theta_1, \ldots, \Theta_\ell\}$. While this approach may seem reasonable, it comes with a drawback. Due to the forward propagation in the GNN, the later losses also depend on the previous parameters $\Theta_{1\ldots\ell}$. However, it is not well studied to what extent they influence the previous parameters in the backward pass. Vanishing gradients could potentially lead to diminishing influence and a separation into layer-wise independent losses $\text{L}_{\Theta_1}, \text{L}_{\Theta_2}, \ldots, \text{L}_{\Theta_\ell}$ that are effectively optimised independently. We show in Appendix B that such an independent optimisation can lead to solutions where $\text{L}_{\Theta_1}$ is minimised first, restricting the solution space for $\text{L}_{\Theta_2}$ such that the cumulative loss is not minimal despite the first pooling operator's optimality. For a better result, the first pooling step must sacrifice optimality such that the second operator has a larger solution space, leading to a lower cumulative loss. We address this issue by explicitly modelling the interdependencies between different hierarchical levels in a single loss $\text{L}_\Theta^{(\ell)} = \text{L}_{\Theta_{1\ldots\ell}}$, enabling us to jointly optimise pooling across all levels.

**Adaptive Depth via Minimum Description Length:** Previous works only consider a fixed hierarchical depth, which is misaligned with empirical graph datasets that can contain graphs of various depths. Our adaptive multilevel pooling architecture (see Figure 1(b)) learns pooling operators for different depths in parallel, up to the maximum depth $\ell$. Thanks to the minimum description length principle, the losses for different depths are comparable, and we can select the best depth for each graph based on the unsupervised clustering loss.

$$\text{MDL-POOL}(\mathbf{A}, \mathbf{H}) := \begin{cases} \text{POOL}_1^{(1)}(\mathbf{A}, \mathbf{H}), & \text{if } \text{L}_\Theta^{(1)} < \text{L}_\Theta^{(l)} \, \forall l \in [1, \ell] \setminus \{1\} \\ \text{POOL}_2^{(2)}(\mathbf{A}, \mathbf{H}), & \text{if } \text{L}_\Theta^{(2)} < \text{L}_\Theta^{(l)} \, \forall l \in [1, \ell] \setminus \{2\} \\ \dots \end{cases} \quad (6)$$

**Downstream task:** To solve a downstream graph-level task, we preprocess the node features $\mathbf{X}$ and adjacency $\mathbf{A}$ with a GNN, creating an embedding $\mathbf{H}$, which we feed into our multilevel pooling architecture. We then compute graph-level embedding vectors $\mathbf{Y}$ by applying a GNN to the multilevel pooling operator's output, followed by a readout function, we use mean pooling, followed by an MLP.

$$Y := \text{MLP}(\text{READOUT}(\text{GNN}(\text{MDL-POOL}(\text{GNN}(\mathbf{A}, \mathbf{X}))))) \quad (7)$$

The overall loss $\text{L}_\Theta$ is a sum of the downstream task's classification loss $\text{L}_\Theta^c$ and the pooling losses, $\text{L}_\Theta := \text{L}_\Theta^c(Y, \hat{Y}) + \sum_{k=1}^\ell \text{L}_\Theta^{(k)}$, such that we optimise all pooling operators at the same time.

## 4.2 OPTIMISATION WITH THE MULTILEVEL MAP EQUATION

We adapt the multilevel map equation as a clustering objective because it satisfies all the desired properties discussed before. Building on network flow, it produces interpretable communities and models the interdependencies between different hierarchical levels. Because it builds on the MDL principle, we can directly compare clusterings with different depths and choose the optimal number of communities and levels. Starting from Equation (1), we derive a hierarchical clustering objective and optimise it indirectly via the soft cluster assignments $\mathbf{S} = \text{softmax}(\text{MLP}_\Theta(\mathbf{A}, \mathbf{X}))$ through the MLP's parameters $\Theta$. In the remainder, we fix the number of levels to $\ell = 2$, however, our loss formulation and experimental setup can easily be expanded to an arbitrary depth. We provide detailed derivations and generalisations for arbitrary depth in Appendix A.2.

For $\ell = 2$ we learn two soft cluster assignment matrices $\mathbf{S}_2^1 \in \mathbb{R}^{n \times m}$ and $\mathbf{S}_2^2 \in \mathbb{R}^{m \times M}$ that pool the $n = |V|$ nodes into at most $m$ sub-modules and these $m$ sub-modules further into at most $M$ modules. Consider a weighted graph, where $w_{uv}$ is the weight of link $(u, v)$, $w_u = \sum_{v \in V} w_{uv}$ is node $u$'s total weight, and $w_{\text{tot}} = \sum_{u \in V} \sum_{v \in V} w_{uv}$ is the total weight in the graph. We denote the graph's transition matrix as $\mathbf{T}_{uv} := w_{uv}/w_u$, and node $u$'s visit rate as $\mathbf{p}_u = w_u/w_{\text{tot}}$. The flow matrix $\mathbf{F}$ encodes the flow between each pair of nodes and is computed as $\mathbf{F}_{uv} = \mathbf{p}_u \mathbf{T}_{uv}$ in undirected networks. In directed networks, we use smart teleportation to compute $\mathbf{F}$ and $\mathbf{p}$ (see Appendix A) (Lambiotte & Rosvall, 2012). The flow $\mathbf{C}_m := \mathbf{S}_2^{1\top} \mathbf{F} \mathbf{S}_2^1$ between clusters, derived from the sub-module assignments $\mathbf{S}_2^1$, is pooled from the flow matrix $\mathbf{F}$. The flow $\mathbf{C}_M := \mathbf{S}_2^{2\top} \mathbf{C}_m \mathbf{S}_2^2 = \mathbf{S}_2^{2\top} \mathbf{S}_2^{1\top} \mathbf{F} \mathbf{S}_2^1 \mathbf{S}_2^2$ for the top-level modules is obtained by pooling from $\mathbf{C}_m$. For larger $\ell$, such a pooling step is done at each intermediate level. We obtain the module entry rates, $\mathbf{q}_M$ and $\mathbf{q}_m$, and exit rates, $\mathbf{M}_{\text{exit}}$ and $\mathbf{m}_{\text{exit}}$, from the cluster flow matrices $\mathbf{C}_M$ and $\mathbf{C}_m$, respectively, and the rate for entering modules at the highest level as $q = 1 - \text{Tr}(\mathbf{C}_M)$.

$$\mathbf{q}_m = \mathbf{C}_m^\top \mathbf{1}_{|m|} - \text{diag}(\mathbf{C}_m) \qquad \mathbf{m}_{\text{exit}} = \mathbf{C}_m \mathbf{1}_{|m|} - \text{diag}(\mathbf{C}_m) \qquad \mathbf{p}_m = \mathbf{m}_{\text{exit}} + \mathbf{S}_2^{1\top} \mathbf{p}_u \quad (8)$$

$$\mathbf{q}_M = \mathbf{C}_M^\top \mathbf{1}_{|M|} - \text{diag}(\mathbf{C}_M) \qquad \mathbf{M}_{\text{exit}} = \mathbf{C}_M \mathbf{1}_{|M|} - \text{diag}(\mathbf{C}_M) \qquad \mathbf{p}_M = \mathbf{q}_M + \mathbf{S}_2^{2\top} \mathbf{q}_m \quad (9)$$

Finally, we obtain the multilevel loss for two pooling layers, $\text{L}_\Theta^{(2)} = \mathcal{L}_2(\mathbf{A}, \mathbf{S}_2^1, \mathbf{S}_2^2)$, where

$$\mathcal{L}_2(\mathbf{A}, \mathbf{S}_2^1, \mathbf{S}_2^2) = q \log_2 q + \sum_{j \in M} [\mathbf{p}_M \log_2 \mathbf{p}_M - \mathbf{q}_M \log_2 \mathbf{q}_M - \mathbf{M}_{\text{exit}} \log_2 \mathbf{M}_{\text{exit}}]_j \quad (10)$$

$$+ \sum_{i \in m, j \in M} [\mathbf{p}_m \log_2 \mathbf{p}_m - \mathbf{q}_m \log_2 \mathbf{q}_m - \mathbf{m}_{\text{exit}} \log_2 \mathbf{m}_{\text{exit}}]_{ij} + \sum_{u \in V} [-\mathbf{p} \log_2 \mathbf{p}]_u \quad (11)$$

with logarithms applied component-wise. Different from other deep clustering methods and thanks to the MDL principle, the map equation loss does not require regularisation to prevent trivial solutions (Blöcker et al., 2024). We call our pooling operator *Minimum Description Length Pooling (MDL-Pool)* and learn it by optimising the multilevel loss $\mathcal{L}_2(\mathbf{A}, \mathbf{S}_2^1, \mathbf{S}_2^2)$ with the model architecture shown in Figure 1, which can be easily adjusted for $\ell > 2$. We discuss MDL-Pool's complexity in Appendix C.

## 5 EXPERIMENTAL EVALUATION

We evaluate MDL-Pool against nine deep graph clustering and pooling baselines on community detection and graph classification tasks. For a fair comparison, we use the same base GNN for all pooling operators, that is, a GIN with two layers and 64 channels, which has been shown to be effective for graph classification tasks (Xu et al., 2019). While deeper GINs or other base GNNs may provide better performance, we focus on isolating and comparing the effects of different pooling methods rather than achieving the best possible GNN performance.

The datasets we use in our experiments vary in size, ranging from small to large. For community detection, we use graphs with up to 19,717 nodes. For graph classification, datasets include up to 41,127 graphs, some of them with an average of up to 430 nodes. We provide further details about the base model, training procedures, datasets, and links to our code for reproducibility in Appendix D.

### 5.1 COMMUNITY DETECTION

We evaluate MDL-Pool against soft-clustering-based pooling methods, BN-Pool (Castellana & Bianchi, 2025), DiffPool (Ying et al., 2018), MinCut (Bianchi et al., 2020), Deep Modularity Network (DMoN) (Tsitsulin et al., 2023), and Just-Balance Graph Neural Network (JBGNN) (Bianchi, 2023), on two synthetic datasets (`Community` and `SBM`) generated from a stochastic block model, and four real-world citation networks (`CiteSeer`, `DBLP`, `Cora`, and `PubMed`).

Community detection is an unsupervised task that traditionally relies solely on the graph's topology to identify clusters in a graph; with GNNs, deep community detection approaches naturally incorporate (node) features into the process. Our setup for deep community detection uses a stack of message-passing layers followed by a pooling operator that transforms the learnt embeddings into a soft cluster assignment matrix $\mathbf{S}$ with at most $c_{\max}$ communities, where $c_{\max}$ is a parameter. To evaluate the goodness of the detected communities, we compare them against the ground truth. However, we note that, for empirical datasets, the ground-truth communities may be difficult, if not impossible, to obtain because the precise data generation process is generally unknown (Peel et al., 2017).

We use Normalised Mutual Information (NMI) (Romano et al., 2014) to measure the alignment between detected and ground truth communities, and show the results in Table 2 (results for Overlapping NMI (ONMI) (McDaid et al., 2013) in Appendix E). For NMI, we use the $\arg\max$ of the soft cluster assignments $\mathbf{S}$ to select the most prominent cluster per node. To reduce noise that may arise from incomplete convergence, we discard assignments whose value is at most $1/c_{\max}$. To test which methods accurately infer the number of clusters, we set $c_{\max} \in \{10, 20, 30, 40, 50\}$.

Figure 2 shows that MDL-Pool, followed by JBGNN and DMoN, shows the most stable performance in terms of the returned number of communities. DiffPool and BN-Pool tend to return more clusters as we increase $c_{\max}$, and MinCut tends to distribute all nodes equally across all clusters, essentially returning a single community. DMoN collapses to a trivial solution on the `SBM` dataset.

The NMI results in Table 2 show that all methods, except for MDL-Pool, perform better when we set $c_{\max}$ to the "correct number" of clusters, $|C|$, which limits their applicability in practical scenarios where this information is not available. We attribute MDL-Pool's slightly worse performance when setting $c_{\max} = |C|$ to the restricted flexibility during training. When the "ground truth" number of clusters is not provided, MDL-Pool outperforms all baselines in five of the six data sets, demonstrating its ability to infer meaningful clusters without relying on additional hyperparameters.

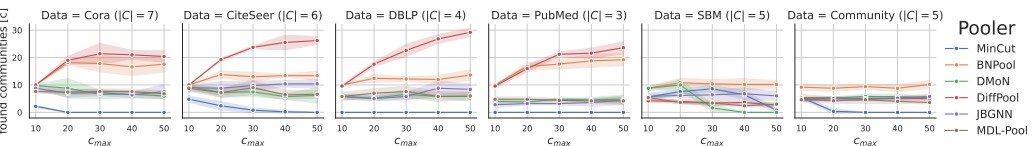

Figure 2: The number of detected communities for setting $c_{\max} \in \{10, 20, 30, 40, 50\}$.

Table 2: Community detection performance of soft-clustering-based pooling methods. (Top) We set the maximum number of clusters to match the ground truth, $c_{max} = |C|$. (Bottom) We consider the number of clusters unknown, setting $c_{max} = 50$. We list the average NMI over 5 runs (ONMI in Appendix E), and the median number of detected communities, $\tilde{c}$; overall best results marked in red.

| | Method | CiteSeer $|C| = 6$ | Community $|C| = 5$ | Cora $|C| = 7$ | DBLP $|C| = 4$ | PubMed $|C| = 3$ | SBM $|C| = 5$ |
|---|---|---|---|---|---|---|---|
| $c_{max} = |C|$ | BNPool | $5.0 \pm 0.7$ (5) | $52.6 \pm 5.7$ (5) | $10.1 \pm 2.0$ (5) | $25.7 \pm 0.4$ (4) | $10.5 \pm 0.3$ (3) | $78.5 \pm 0.5$ (3) |
| | DiffPool | $18.1 \pm 0.5$ (6) | $78.6 \pm 1.1$ (5) | $30.9 \pm 3.9$ (7) | $8.2 \pm 2.6$ (4) | $10.0 \pm 0.9$ (3) | $100.0 \pm 0.0$ (5) |
| | DMoN | $19.5 \pm 5.6$ (6) | $87.8 \pm 5.8$ (5) | $31.5 \pm 3.7$ (7) | $19.7 \pm 5.4$ (4) | $17.5 \pm 5.7$ (3) | $100.0 \pm 0.0$ (5) |
| | JBGNN | $15.0 \pm 5.0$ (6) | $93.8 \pm 1.7$ (5) | $23.7 \pm 4.5$ (7) | $18.9 \pm 4.2$ (4) | $5.4 \pm 5.8$ (3) | $96.5 \pm 4.8$ (5) |
| | MinCut | $19.2 \pm 3.3$ (6) | $89.8 \pm 0.7$ (5) | $37.0 \pm 3.1$ (7) | $32.2 \pm 1.3$ (4) | $17.2 \pm 5.3$ (3) | $100.0 \pm 0.0$ (5) |
| | MDL-Pool | $14.5 \pm 4.2$ (6) | $85.3 \pm 5.0$ (4) | $35.2 \pm 5.4$ (6) | $21.6 \pm 5.9$ (4) | $22.3 \pm 6.3$ (3) | $96.6 \pm 4.6$ (5) |
| $c_{max} = 50$ | BNPool | $5.4 \pm 0.7$ (5) | $44.4 \pm 3.9$ (6) | $8.8 \pm 1.0$ (5) | $22.2 \pm 3.4$ (7) | $8.8 \pm 3.5$ (13) | $59.2 \pm 1.2$ (2) |
| | DiffPool | $20.2 \pm 1.1$ (50) | $63.2 \pm 0.3$ (36) | $34.9 \pm 0.5$ (50) | $13.9 \pm 2.1$ (50) | $13.6 \pm 0.5$ (50) | $100.0 \pm 0.0$ (5) |
| | DMoN | $17.6 \pm 1.0$ (49) | $56.2 \pm 0.4$ (50) | $28.9 \pm 1.2$ (50) | $14.8 \pm 0.9$ (50) | $12.9 \pm 1.8$ (50) | $67.5 \pm 1.1$ (32) |
| | JBGNN | $16.2 \pm 1.2$ (45) | $66.6 \pm 0.5$ (27) | $26.7 \pm 3.4$ (46) | $16.9 \pm 2.6$ (49) | $6.8 \pm 2.2$ (49) | $91.1 \pm 2.1$ (7) |
| | MinCut | $12.7 \pm 6.4$ (37) | $0.0 \pm 0.0$ (1) | $13.4 \pm 1.8$ (35) | $2.6 \pm 0.8$ (8) | $0.6 \pm 0.5$ (8) | $99.2 \pm 1.7$ (5) |
| | MDL-Pool | $16.3 \pm 1.4$ (12) | $96.9 \pm 0.0$ (5) | $37.1 \pm 3.1$ (11) | $26.0 \pm 2.0$ (11) | $23.2 \pm 3.4$ (10) | $100.0 \pm 0.0$ (5) |

## 5.2 GRAPH CLASSIFICATION

Graph classification involves assigning a class label to a graph based on its structure and features. We use the same setup as for community detection, but add a message-passing layer after pooling and a readout function to obtain a graph-level representation (see Figure 1 and Equation (7)). The model is trained by minimising the sum of the supervised cross-entropy classification and the unsupervised pooling losses. In addition to the clustering-based methods, we include the 1-Every-K and score-based pooling methods Top-k (Gao & Ji, 2019), EdgeContraction Pooling (ECPool) (Diehl, 2019), k Maximal Independent Sets Pooling (k-MIS) (Bacciu et al., 2023), and Graclus (Dhillon et al., 2007); these methods do not involve any pooling loss and are trained solely using the supervised classification loss. Furthermore, we include a baseline model without pooling (nopool) for comparison.

Table 3 lists the results on the TUData benchmarks (Morris et al., 2020) and ogb-molhiv (Hu et al., 2020). Overall, there is always a pooling method that outperforms the nopool baseline, highlighting the importance of pooling for graph classification. MDL-Pool achieves state-of-the-art performance on D&D and IMDB-BINARY, and performs competitively with other clustering-based methods despite not requiring hyperparameters for the number of clusters or levels. Figure 3 shows the selected pooling depth for MDL-Pool per dataset. In most cases, a depth of one is chosen, indicating that pooling generally enhances performance. We also tested using a depth of up to three but this did not improve the performance (see Appendix F). There is no one-depth-fits-all setting for any dataset, highlighting the importance of adaptively selecting the best depth per graph instance. For larger graphs, we expect the importance of an adaptive selection process to become even more apparent.

Table 3: Empirical classification results (ACC) for one-every-K- or score-based, clustering-based and parameter-free clustering-based pooling methods. Appendix D provides experiment details and code.

| | Pooler | COLLAB | COLORS-3 | D&D | ENZYMES | IMDB-B | MUTAG | Mutag. | NCI1 | PROTEINS | REDDIT-B | molhiv (AUROC) |
|---|---|---|---|---|---|---|---|---|---|---|---|---|
| | nopool | $75.8 \pm 1.4$ | $93.4 \pm 2.3$ | $75.1 \pm 2.7$ | $41.7 \pm 5.1$ | $75.6 \pm 6.2$ | $87.1 \pm 3.2$ | $81.1 \pm 1.5$ | $79.6 \pm 2.3$ | $75.9 \pm 7.0$ | $92.0 \pm 1.8$ | $75.8 \pm 2.5$ |
| Score, 1/K | ECPool | $77.0 \pm 1.4$ | $82.3 \pm 2.6$ | $75.3 \pm 1.8$ | $42.3 \pm 5.3$ | $76.4 \pm 10.9$ | $87.1 \pm 3.2$ | $81.4 \pm 2.2$ | $80.6 \pm 2.1$ | $74.7 \pm 6.3$ | $93.0 \pm 1.0$ | $77.4 \pm 1.0$ |
| | Graclus | $77.1 \pm 1.6$ | $83.5 \pm 2.4$ | $71.4 \pm 1.9$ | $42.7 \pm 6.8$ | $74.8 \pm 8.1$ | $85.7 \pm 8.7$ | $82.3 \pm 1.8$ | $79.4 \pm 1.5$ | $75.5 \pm 5.1$ | $92.5 \pm 0.9$ | $77.1 \pm 1.2$ |
| | k-MIS | $74.9 \pm 1.4$ | $92.2 \pm 1.1$ | $75.6 \pm 1.4$ | $40.7 \pm 8.5$ | $74.8 \pm 7.3$ | $88.6 \pm 6.4$ | $80.8 \pm 1.6$ | $80.1 \pm 1.4$ | $76.5 \pm 4.9$ | $92.0 \pm 2.4$ | $75.4 \pm 2.6$ |
| | Top-$k$ | $74.3 \pm 1.8$ | $77.2 \pm 17.0$ | $72.4 \pm 4.3$ | $39.7 \pm 3.6$ | $74.4 \pm 11.6$ | $87.1 \pm 9.3$ | $78.0 \pm 1.4$ | $77.7 \pm 2.1$ | $73.3 \pm 4.9$ | $91.0 \pm 0.5$ | $75.6 \pm 2.9$ |
| Clustering | DiffPool | $60.8 \pm 1.9$ | $76.8 \pm 6.2$ | $62.0 \pm 5.3$ | $16.3 \pm 4.3$ | $72.0 \pm 8.7$ | $87.1 \pm 9.3$ | $78.6 \pm 1.9$ | $70.4 \pm 9.3$ | $75.5 \pm 4.5$ | $80.5 \pm 10.1$ | $73.3 \pm 3.2$ |
| | DMoN | $76.0 \pm 0.9$ | $90.9 \pm 0.9$ | $77.1 \pm 3.8$ | $42.7 \pm 5.5$ | $74.8 \pm 4.6$ | $90.0 \pm 6.4$ | $80.8 \pm 1.7$ | $80.2 \pm 2.7$ | $76.5 \pm 4.7$ | $91.1 \pm 1.1$ | $74.9 \pm 0.8$ |
| | JBGNN | $75.7 \pm 1.2$ | $89.0 \pm 4.0$ | $77.3 \pm 4.3$ | $45.0 \pm 6.8$ | $76.8 \pm 7.7$ | $87.1 \pm 9.3$ | $81.6 \pm 1.2$ | $79.3 \pm 1.9$ | $77.1 \pm 3.9$ | $91.8 \pm 1.2$ | $75.9 \pm 2.1$ |
| | MinCut | $75.8 \pm 1.4$ | $91.8 \pm 1.4$ | $78.3 \pm 2.8$ | $41.3 \pm 5.9$ | $73.6 \pm 6.5$ | $87.1 \pm 7.8$ | $81.2 \pm 0.9$ | $80.0 \pm 0.7$ | $76.1 \pm 5.4$ | $91.6 \pm 1.5$ | $76.5 \pm 1.5$ |
| Free | BNPool | $73.5 \pm 0.7$ | $97.1 \pm 0.7$ | $74.7 \pm 3.7$ | $38.0 \pm 3.6$ | $75.6 \pm 6.7$ | $85.7 \pm 5.1$ | $80.1 \pm 1.9$ | $78.6 \pm 1.4$ | $76.3 \pm 3.6$ | $90.4 \pm 2.0$ | $76.8 \pm 2.1$ |
| | MDL-Pool (1-LVL) | $68.9 \pm 6.0$ | $86.5 \pm 1.2$ | $77.3 \pm 2.0$ | $41.3 \pm 5.2$ | $76.0 \pm 5.1$ | $90.0 \pm 8.1$ | $80.5 \pm 0.8$ | $78.0 \pm 1.7$ | $75.9 \pm 4.6$ | $91.3 \pm 1.8$ | $76.3 \pm 1.0$ |
| | MDL-Pool | $76.3 \pm 0.9$ | $87.2 \pm 1.8$ | $79.7 \pm 2.5$ | $39.3 \pm 3.2$ | $77.2 \pm 5.4$ | $85.7 \pm 8.7$ | $80.0 \pm 2.0$ | $79.0 \pm 1.2$ | $76.1 \pm 5.5$ | $91.6 \pm 1.1$ | $75.2 \pm 2.0$ |

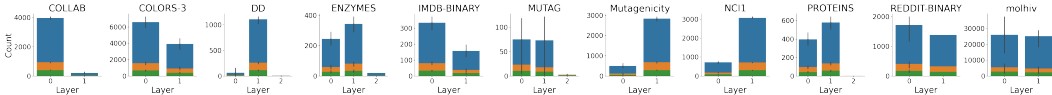

Figure 3: Distribution of number of pooling layers for graphs (train, val, test) selected by MDL-Pool. We tested a pooling depth of up to three for all datasets, but the maximum chosen depth was two.

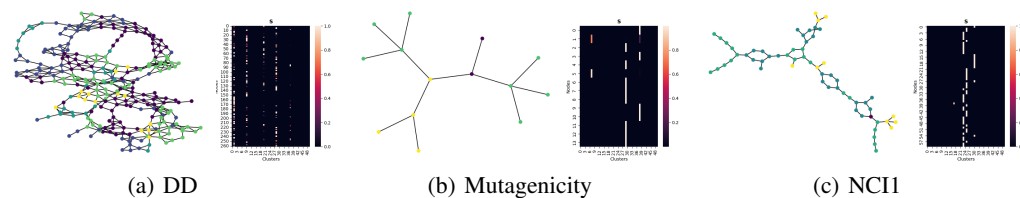

|  (a) DD | (b) Mutagenicity | (c) NCI1 |

Figure 4: MDL-Pool tends to learn clean node assignment with little overlap between clusters.

In Figure 4, we show the learnt cluster assignments for selected graphs and find that, despite its ability to return soft clusters, MDL-Pool rarely assigns nodes to multiple clusters. Notably, MDL-Pool identifies recurring substructures within the graphs, which are often sensible and meaningful. According to the map equation, disconnected structures should preferably be assigned to different modules, however, since the pooling operator relies on the combination of embeddings and topology, such structures are not separated when they are highly similar, for example, due to their features. This effect is particularly evident in the Mutagenicity example.

### 5.3 ABLATION STUDY AND LIMITATIONS

**Ablation Study.** We investigate which design choices in our setup are essential for MDL-Pool: We compare adaptive pooling with fixed-depth pooling, test depth limits, and evaluate the effect of treating partially assigned nodes and singleton clusters differently. Moreover, we employ a GIN instead of MLP in the SEL operation and assess the effect of isolating non-selected pooling operators from influencing the pre-layer. We report the results in Appendix F.

**Limitations.** We followed the setup proposed by Castellana & Bianchi (2025), however, we did not perform model-specific hyperparameter tuning, which could improve the performance of some methods further. We assume connected graphs; in the case of disconnected graphs, pooling should be applied for each component separately. MDL-Pool only uses the graph's topology to measure cluster quality via the map equation, but not the features, which may be a useful extension we leave for future work. Furthermore, we directly use the datasets' predefined features, while more advanced features could enable our method to split communities with similar nodes, leading to better performance.

### 6 CONCLUSION

We proposed *MDL-Pool*, an adaptive multilevel graph pooling operator based on the map equation. MDL-Pool satisfies desirable properties for pooling operators that we identified: It learns interpretable hierarchical clusters and automatically determines the optimal number of clusters and pooling depth from the data. Different from previous works, our multilevel pooling objective function jointly optimises the clusters across multiple levels instead of merely stacking shallow clustering operators. MDL-Pool follows the minimum description length principle, making it a parameter-free graph pooling method that does not require explicit regularisation, which was essential in previous works.

In an empirical evaluation on eleven common graph classification and six community detection datasets, *MDL-Pool* performs competitively against the baselines, returning more accurate communities than the baselines in five out of six cases. In graph classification, MDL-Pool achieves state-of-the-art performance in two of the eleven scenarios. However, in line with no-free-lunch theorems for optimisation (Wolpert & Macready, 1997) and community detection (Peel et al., 2017), as well as previous works on graph pooling, we do not find a clear winner for graph classification.

Our work raises some open questions for future work: Current clustering methods necessarily merge communities that are distributed across the graph if their nodes share similar features. While these clusters remain meaningful, the map equation would typically separate them because disconnected communities increase the codelength. A possible way to address this is to design features based on the graph's topology to facilitate splitting such communities. Alternatively, more expressive embeddings may allow us to distinguish between such modules.

## ETHICS STATEMENT

We have read the ICLR Code of Ethics and confirm that, to the best of our knowledge, we adhere to it. In this work, we focus on basic machine learning research and propose a novel graph pooling method and do not expect negative societal impacts that go beyond those of other foundational works in machine learning research.

## REPRODUCIBILITY STATEMENT

We have described our proposed graph pooling method in detail in Section 4, and our experimental evaluation setup in Section 5 and Appendix D. In case our paper is accepted, we will make a GitHub repository with our codebase available. All used datasets are open source and freely available online.

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

## A    MAP EQUATION DETAILS

The map equation is an information-theoretic objective function for community detection that builds on the minimum description length (MDL) principle (Rosvall et al., 2009; Smiljanić et al., 2023). It uses random walks on networks as a proxy for the network's structure and, given a partition of the nodes into modules, computes the expected number of bits required to describe a random walker step on the network—the *codelength*. However, as always in information theory, we are not interested in concrete codewords for describing random walks (Grünwald et al., 2005). Instead, we care about the theoretical codelength for a given network partition. Nevertheless, discussing the map equation in terms of concrete random walks and codewords is a useful way to explain its inner workings.

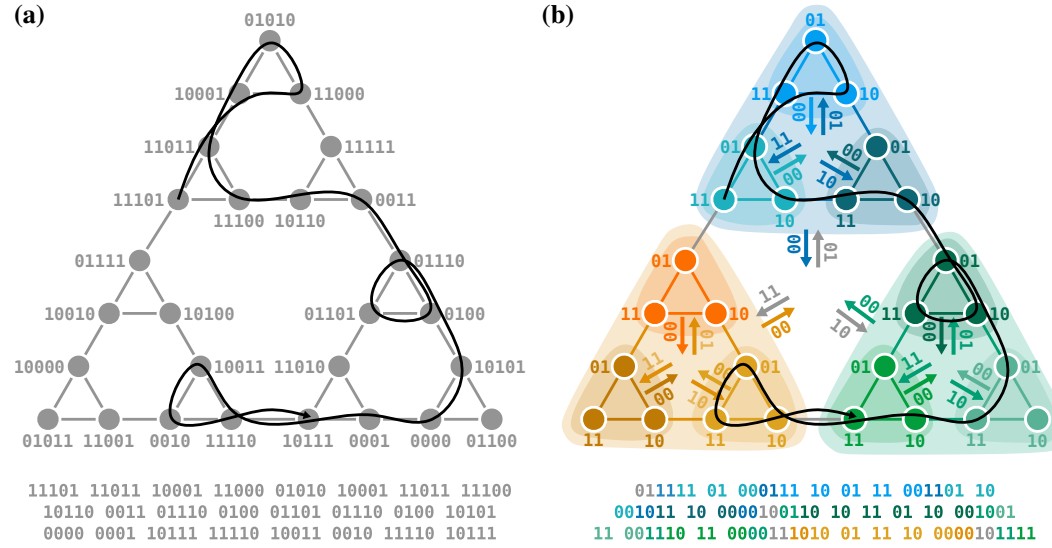

Figure 5: Coding principles behind the map equation. **(a)** All nodes are assigned to the same module and receive unique codewords, constructed with a Huffman code (Huffman, 1952) based on their ergodic visit rates. The black trace shows a possible sequence of node visits by a random walker. For each step, we use one codeword, resulting in the sequence of codewords shown at the bottom. **(b)** The nodes are partitioned into nested modules where colours show module memberships. With modules, we assign unique codewords within modules, but we must also define codewords for describing module entries and exits. These are shown next to the coloured arrows pointing into and out of modules. Now, a random walker step requires one, three, or five codewords, depending on how many module boundaries are crossed. With these codes, the codelength for the same sequence of nodes becomes shorter, as shown at the bottom.

## A.1 How the Map Equation Works

Let $G = (V, E)$ be a graph with nodes $V$ and links $E$, possibly directed, and let $w_{uv}$ be the weight of link $(u, v)$ with $u, v \in V$. In the simplest case, there are no modules as shown in Figure 5(a). The map equation literature refers to this case as the so-called "one-level partition", but because we think about it as applying zero pooling steps, we use a slightly different naming convention. In this case, nodes receive unique codewords based on their ergodic visit rates, constructed via Huffman coding (Huffman, 1952). According to Shannon's source coding theorem (Shannon, 1948), the average required number of bits to describe a random walk is the entropy of the nodes' ergodic visit rates, that is $\mathcal{H}(P) = \sum_{u \in V} p_u \log_2 p_u$, where $P = \{p_u \mid u \in V\}$ is the set of ergodic node visit rates, and $\mathcal{H}$ is the Shannon entropy.

Node $u$'s visit rates can be computed in closed form as $p_u = \sum_v w_{uv} / \sum_u \sum_v w_{uv}$ in undirected graphs. In strongly connected directed graphs, it can be computed with a power iteration to solve the set of equations $p_v = \sum_u p_u \mathbf{T}_{uv}$, where

$$\mathbf{T}_{uv} = \begin{cases} \dfrac{w_{uv}}{\sum_{v \in V} w_{uv}} & \text{if } \sum_{v \in V} w_{uv} > 0, \\ 0 & \text{otherwise}, \end{cases} \tag{12}$$

are the entries of the random walkers's transition matrix $\mathbf{T}$ (Smiljanić et al., 2023). In weakly connected graphs, PageRank (Gleich, 2015) or smart teleportation (Lambiotte & Rosvall, 2012) can be used to obtain the visit rates. PageRank, which uses uniform node teleportation to ensure a strongly-connected graph, computes the nodes' visit rates with a power iteration and the following update rule and initial values,

$$p_v^{(i+1)} \leftarrow \alpha \cdot \frac{1}{|V|} + (1 - \alpha) \cdot \sum_{u \in V} p_u^{(i)} \mathbf{T}_{uv}, \qquad p_v^{(0)} \leftarrow \frac{1}{|V|}, \tag{13}$$

where $\alpha \in [0, 1]$ is the random walker's teleportation rate. That is, in each step, the random walker follows a link with probability $1 - \alpha$, or teleports to a randomly chosen node with probability $\alpha$. Smart teleportation instead employs uniform link teleportation and a power iteration with the following update rule and initial values,

$$p_v^{(i+1)} \leftarrow \alpha \cdot \frac{\sum_{u \in V} w_{uv}}{w_{\text{tot}}} + (1 - \alpha) \sum_{u \in V} p_u^{(i)} \mathbf{T}_{uv}, \qquad p_v^{(0)} \leftarrow \frac{\sum_{u \in V} w_{uv}}{w_{\text{tot}}}, \tag{14}$$

where $w_{\text{tot}} = \sum_{u, v \in V} w_{uv}$ is the total weight in the graph. That is, in each step, the random walker follows a link with probability $1 - \alpha$, or teleports to a node with probability $\alpha$, where the nodes are chosen proportionally to their in-degree.

Many graphs have communities, and reflecting those communities in how the nodes are partitioned into modules can reduce the codelength. Shorter codewords become possible because we assign unique codewords within modules, but we reuse the same codewords for nodes in different modules. However, for a uniquely decodable code, we must introduce a so-called index-level codebook for encoding transitions between modules. The standard map equation generalises Shannon entropy for partitions of nodes into modules and computes the codelength for a given non-hierarchical partition M as

$$\mathcal{L}_1(\mathsf{M}) = q\mathcal{H}(Q) + \sum_{\mathsf{m} \in \mathsf{M}} p_{\mathsf{m}} \mathcal{H}(P_{\mathsf{m}}). \tag{15}$$

That is, the codelength is a weighted average of module-level entropies, one per module, plus an additional weighted entropy term for the so-called index level for describing transitions between modules. Here, $q = \sum_{\mathsf{m} \in \mathsf{M}} q_{\mathsf{m}}$ is the index codebook usage rate, $q_{\mathsf{m}} = \sum_{u \notin \mathsf{m}} \sum_{v \in \mathsf{m}} p_u \mathbf{T}_{uv}$ is module m's entry rate, and $Q = \{q_{\mathsf{m}}/q \mid \mathsf{m} \in \mathsf{M}\}$ is the set of normalised module entry rates. $p_{\mathsf{m}} = \mathsf{m}_{\text{exit}} + \sum_{u \in \mathsf{m}} p_u$ is the codebook usage rate for module m, $\mathsf{m}_{\text{exit}} = \sum_{u \in \mathsf{m}} \sum_{v \notin \mathsf{m}} p_u \mathbf{T}_{uv}$ is module m's exit rate, and $P_{\mathsf{m}} = \{\mathsf{m}_{\text{exit}}/p_{\mathsf{m}}\} \cup \{p_u/p_{\mathsf{m}} \mid u \in \mathsf{m}\}$ is the set of normalised node visit rates and exit rate for module m. With such a coding scheme, we use one codeword for describing random walker steps within modules, or three codewords when the random walker changes modules. With our zero-based naming convention, $\mathcal{L}_1$ reflects that one step of pooling will be applied based on the given communities M, whereas the map equation literature refers to this case as a "two-level partition".

The multilevel map equation (Rosvall & Bergstrom, 2011; Smiljanić et al., 2023) uses recursion to generalise the map equation to hierarchical partitions,

$$\mathcal{L}_\ell\left(\mathsf{M}\right) = q\mathcal{H}\left(Q\right) + \sum_{\mathsf{m} \in \mathsf{M}} \mathcal{L}_{\ell-1}\left(\mathsf{m}\right), \tag{16}$$

where $\mathsf{M}$ is an $\ell$-level partition of the nodes into modules. Figure 5(b) shows an example where the nodes are partitioned into nested modules in two levels.

To implement the map equation in matrix form, we rely on the flow matrix $\mathbf{F}$, which captures the amount of flow between each pair of nodes. It is computed as $\mathbf{F}_{uv} = \mathbf{p}_u \mathbf{T}_{uv}$ in undirected graphs. In directed graphs, we use smart teleportation and a power iteration according to Equation (14) to first compute the nodes' visit rates $\mathbf{p}$ with the following update rule (Lambiotte & Rosvall, 2012),

$$\mathbf{p}^{(t+1)} \leftarrow \alpha\mathbf{d}^{\text{in}} + \left(1 - \alpha\right)\mathbf{p}^{(t)}\mathbf{T} \qquad \mathbf{p}^{(0)} = \mathbf{d}^{\text{in}} \qquad \mathbf{d}_v^{\text{in}} = \frac{\sum_{u \in V} w_{uv}}{w_{\text{tot}}}, \tag{17}$$

where $\mathbf{d}_v^{\text{in}}$ is node $v$'s in-strength and $\alpha$ is a teleporation parameter. Then, we compute the flow matrix $\mathbf{F}$ as (Blöcker et al., 2024)

$$\mathbf{F} = \frac{\alpha}{w_{\text{tot}}}\mathbf{A} + \left(1 - \alpha\right)\text{diag}\left(\mathbf{p}\right)\mathbf{T}. \tag{18}$$

### A.2 Expansion of the $\ell$-level Map Equation

In the following, we show how we obtain the expanded $\ell$-level map equation (Equation (43)) from its recursive definition (Equation (1)). For simplicity, we begin with $\ell = 2$. Note that, for clarity and to match the number of applied pooling operations, we adopt a zero-based naming convention for the map equation. The original map equation work by Rosvall et al. (2009) refers to the case without communities as the "one-level partition" and to cases with a single level of communities as "two-level" partitions. That is, the $\ell$-level map equation in our case corresponds to $(\ell + 1)$-level partitions in Rosvall et al. (2009).

#### A.2.1 The Expanded 2-level Map Equation

We denote the top-level modules as $\mathbb{M}$, the modules at the middle level as $\mathsf{M} \in \mathbb{M}$, and the modules at the bottom level as $\mathsf{m} \in \mathsf{M}$. We start with the definition for $\ell = 2$,

$$\mathcal{L}_3(\mathbb{M}) = q\mathcal{H}(Q) + \sum_{\mathsf{M} \in \mathbb{M}}\left[p_{\mathsf{M}}\mathcal{H}\left(P_{\mathsf{M}}\right) + \sum_{\mathsf{m} \in \mathsf{M}} p_{\mathsf{m}}\mathcal{H}\left(P_{\mathsf{m}}\right)\right] \tag{19}$$

To calculate the module-level entropies, we need to distinguish between three cases: (1) modules at the highest level, (2) modules at intermediate levels, and (3) modules at the bottom level.

First, the codelength contribution for transitions between top-level modules is (Smiljanić et al., 2023)

$$\mathcal{H}\left(Q\right) = -\sum_{\mathsf{M} \in \mathbb{M}} \frac{q_{\mathsf{M}}}{q} \log_2 \frac{q_{\mathsf{M}}}{q}, \tag{20}$$

where $q_{\mathsf{M}}$ is module $\mathsf{M}$'s entry rate and $q = \sum_{\mathsf{M} \in \mathbb{M}} q_{\mathsf{M}}$ is the rate at which the index level module is used. Second, at intermediate levels, we need to consider entering sub-modules and exiting to the super-module.

$$\mathcal{H}\left(P_{\mathsf{M}}\right) = -\frac{\mathsf{M}_{\text{exit}}}{p_{\mathsf{M}}}\log_2\frac{\mathsf{M}_{\text{exit}}}{p_{\mathsf{M}}} - \sum_{\mathsf{m} \in \mathsf{M}}\frac{q_{\mathsf{m}}}{p_{\mathsf{M}}}\log_2\frac{q_{\mathsf{m}}}{p_{\mathsf{M}}} \tag{21}$$

In the case of arbitrary $\ell$, the modules at all intermediate levels are treated like this. Third, at the lowest level, modules do not have any further submodules and contain nodes.

$$\mathcal{H}\left(P_{\mathsf{m}}\right) = -\frac{\mathsf{m}_{\text{exit}}}{p_{\mathsf{m}}}\log_2\frac{\mathsf{m}_{\text{exit}}}{p_{\mathsf{m}}} - \sum_{u \in \mathsf{m}}\frac{p_u}{p_{\mathsf{m}}}\log_2\frac{p_u}{p_{\mathsf{m}}} \tag{22}$$

We expand Equation (19) by substituting the definitions from Equations (20) to (22).

$$\mathcal{L}_2\left(\mathbb{M}\right) = -\not{q}\sum_{\mathsf{M}\in\mathbb{M}}\frac{q_\mathsf{M}}{\not{q}}\log_2\frac{q_\mathsf{M}}{q} \tag{23}$$

$$-\sum_{\mathsf{M}\in\mathbb{M}}\not{p_\mathsf{M}}\left(\frac{\mathsf{M}_{\text{exit}}}{\not{p_\mathsf{M}}}\log_2\frac{\mathsf{M}_{\text{exit}}}{p_\mathsf{M}}+\sum_{\mathsf{m}\in\mathsf{M}}\frac{q_\mathsf{m}}{\not{p_\mathsf{M}}}\log_2\frac{q_\mathsf{m}}{p_\mathsf{M}}\right) \tag{24}$$

$$-\sum_{\mathsf{M}\in\mathbb{M}}\sum_{\mathsf{m}\in\mathsf{M}}\not{p_\mathsf{m}}\left(\frac{\mathsf{m}_{\text{exit}}}{\not{p_\mathsf{m}}}\log_2\frac{\mathsf{m}_{\text{exit}}}{p_\mathsf{m}}+\sum_{u\in\mathsf{m}}\frac{p_u}{\not{p_\mathsf{m}}}\log_2\frac{p_u}{p_\mathsf{m}}\right) \tag{25}$$

After simplification with logarithm rules, we obtain

$$\mathcal{L}_2\left(\mathbb{M}\right) = -\sum_{\mathsf{M}\in\mathbb{M}}q_\mathsf{M}\log_2 q_\mathsf{M}+\sum_{\mathsf{M}\in\mathbb{M}}q_\mathsf{M}\log_2 q \tag{26}$$

$$-\sum_{\mathsf{M}\in\mathbb{M}}\mathsf{M}_{\text{exit}}\log_2\mathsf{M}_{\text{exit}}+\sum_{\mathsf{M}\in\mathbb{M}}\mathsf{M}_{\text{exit}}\log_2 p_\mathsf{M} \tag{27}$$

$$-\sum_{\mathsf{M}\in\mathbb{M}}\sum_{\mathsf{m}\in\mathsf{M}}q_\mathsf{m}\log_2 q_\mathsf{m}+\sum_{\mathsf{M}\in\mathbb{M}}\sum_{\mathsf{m}\in\mathsf{M}}q_\mathsf{m}\log_2 p_\mathsf{M} \tag{28}$$

$$-\sum_{\mathsf{M}\in\mathbb{M}}\sum_{\mathsf{m}\in\mathsf{M}}\mathsf{m}_{\text{exit}}\log_2\mathsf{m}_{\text{exit}}+\sum_{\mathsf{M}\in\mathbb{M}}\sum_{\mathsf{m}\in\mathsf{M}}\mathsf{m}_{\text{exit}}\log_2 p_\mathsf{m} \tag{29}$$

$$-\sum_{\mathsf{M}\in\mathbb{M}}\sum_{\mathsf{m}\in\mathsf{M}}\sum_{u\in\mathsf{m}}p_u\log_2 p_u+\sum_{\mathsf{M}\in\mathbb{M}}\sum_{\mathsf{m}\in\mathsf{M}}\sum_{u\in\mathsf{m}}p_u\log_2 p_\mathsf{m} \tag{30}$$

We use the following definitions for the rates from the main text

$$q=\sum_{\mathsf{M}\in\mathbb{M}}q_\mathsf{M} \qquad p_\mathsf{M}=\mathsf{M}_{\text{exit}}+\sum_{\mathsf{m}\in\mathsf{M}}\mathsf{m}_{\text{enter}} \qquad p_\mathsf{m}=\mathsf{m}_{\text{exit}}+\sum_{u\in\mathsf{m}}p_u \tag{31}$$

to simplify further,

$$\mathcal{L}_2\left(\mathbb{M}\right) = -\sum_{\mathsf{M}\in\mathbb{M}}q_\mathsf{M}\log_2 q_\mathsf{M}+q\log_2 q \tag{32}$$

$$-\sum_{\mathsf{M}\in\mathbb{M}}\mathsf{M}_{\text{exit}}\log_2\mathsf{M}_{\text{exit}}-\sum_{\mathsf{M}\in\mathbb{M}}\sum_{\mathsf{m}\in\mathsf{M}}q_\mathsf{m}\log_2 q_\mathsf{m}+\sum_{\mathsf{M}\in\mathbb{M}}p_\mathsf{M}\log_2 p_\mathsf{M} \tag{33}$$

$$-\sum_{\mathsf{M}\in\mathbb{M}}\sum_{\mathsf{m}\in\mathsf{M}}\mathsf{m}_{\text{exit}}\log_2\mathsf{m}_{\text{exit}}-\sum_{\mathsf{M}\in\mathbb{M}}\sum_{\mathsf{m}\in\mathsf{M}}\sum_{u\in\mathsf{m}}p_u\log_2 p_u+\sum_{\mathsf{M}\in\mathbb{M}}\sum_{\mathsf{m}\in\mathsf{M}}p_\mathsf{m}\log_2 p_\mathsf{m} \tag{34}$$

We reorder and combine the terms to the final expansion of the 2-level map equation:

$$\mathcal{L}_2\left(\mathbb{M}\right) = q\log_2 q \tag{35}$$

$$+\sum_{\mathsf{M}\in\mathbb{M}}\left[p_\mathsf{M}\log_2 p_\mathsf{M}-q_\mathsf{M}\log_2 q_\mathsf{M}-\mathsf{M}_{\text{exit}}\log_2\mathsf{M}_{\text{exit}}\right] \tag{36}$$

$$+\sum_{\mathsf{M}\in\mathbb{M}}\sum_{\mathsf{m}\in\mathsf{M}}\left[p_\mathsf{m}\log_2 p_\mathsf{m}-q_\mathsf{m}\log_2 q_\mathsf{m}-\mathsf{m}_{\text{exit}}\log_2\mathsf{m}_{\text{exit}}\right] \tag{37}$$

$$-\sum_{\mathsf{M}\in\mathbb{M}}\sum_{\mathsf{m}\in\mathsf{M}}\sum_{u\in\mathsf{m}}p_u\log_2 p_u \tag{38}$$

### A.2.2 THE EXPANDED 2-LEVEL MAP EQUATION FOR UNDIRECTED NETWORKS

For undirected networks, we can use the symmetries between module entry and exit rates to simplify the expanded 3-level map equation further, that is, $q_{\mathsf{M}} = \mathsf{M}_{\mathrm{exit}}$ and $q_{\mathsf{m}} = \mathsf{m}_{\mathrm{exit}}$ for all $\mathsf{M}$ and $\mathsf{m}$,

$$\mathcal{L}_2\left(\mathbb{M}\right) = q \log_2 q \tag{39}$$

$$- 2 \sum_{\mathsf{M}\in\mathbb{M}} q_{\mathsf{M}} \log_2 q_{\mathsf{M}} + \sum_{\mathsf{M}\in\mathbb{M}} p_{\mathsf{M}} \log_2 p_{\mathsf{M}} \tag{40}$$

$$- 2 \sum_{\mathsf{M}\in\mathbb{M}} \sum_{\mathsf{m}\in\mathsf{M}} q_{\mathsf{m}} \log_2 q_{\mathsf{m}} + \sum_{\mathsf{M}\in\mathbb{M}} \sum_{\mathsf{m}\in\mathsf{M}} p_{\mathsf{m}} \log_2 p_{\mathsf{m}} \tag{41}$$

$$- \sum_{\mathsf{M}\in\mathbb{M}} \sum_{\mathsf{m}\in\mathsf{M}} \sum_{u\in\mathsf{m}} p_u \log_2 p_u \tag{42}$$

### A.2.3 THE EXPANDED $\ell$-LEVEL MAP EQUATION

Based on the expanded 2-level map equation, we obtain the $\ell$-level map equation. Each intermediate clustering level introduces three terms of the form as found in Equations (36) and (37), leading to the $\ell$-level map equation (with redundant parentheses for visual clarity),

$$\mathrm{L}_\ell\left(\mathcal{M}^\ell\right) = q \log_2 q \tag{43}$$

$$+ \sum_{\mathcal{M}^{\ell-1}\in\mathcal{M}^\ell} \left[\left(p_{\mathcal{M}^{\ell-1}} \log_2 p_{\mathcal{M}^{\ell-1}}\right) - \left(q_{\mathcal{M}^{\ell-1}} \log_2 q_{\mathcal{M}^{\ell-1}}\right) - \left(\mathcal{M}^{\ell-1}_{\mathrm{exit}} \log_2 \mathcal{M}^{\ell-1}_{\mathrm{exit}}\right)\right] \tag{44}$$

$$+ \sum_{\mathcal{M}^{\ell-1}\in\mathcal{M}^\ell} \sum_{\mathcal{M}^{\ell-2}\in\mathcal{M}^{\ell-1}} \left[\left(p_{\mathsf{M}^{\ell-2}} \log_2 p_{\mathsf{M}^{\ell-2}}\right) - \left(q_{\mathcal{M}^{\ell-2}} \log_2 q_{\mathcal{M}^{\ell-2}}\right) - \left(\mathcal{M}^{\ell-2}_{\mathrm{exit}} \log_2 \mathcal{M}^{\ell-2}_{\mathrm{exit}}\right)\right]$$

$$\tag{45}$$

$$+ \cdots \tag{46}$$

$$- \sum_{\mathcal{M}^{\ell-1}\in\mathcal{M}^\ell} \sum_{\mathcal{M}^{\ell-2}\in\mathcal{M}^{\ell-1}} \cdots \sum_{u\in\mathcal{M}^1} p_u \log_2 p_u, \tag{47}$$

where $\mathcal{M}_\ell$ is an $\ell$-level partition of the nodes into modules.

Restricting the $\ell$-level map equation to two levels and assuming undirected networks leads to the two-level map equation proposed by Rosvall et al. (2009) (with slightly different notation),

$$\mathcal{L}_2\left(\mathsf{M}\right) = q \log q - 2 \sum_{\mathsf{m}\in\mathsf{M}} q_{\mathsf{m}} \log_2 q_{\mathsf{m}} + \sum_{\mathsf{m}\in\mathsf{M}} p_{\mathsf{m}} \log_2 p_{\mathsf{m}} - \sum_{u\in V} p_u \log_2 p_u. \tag{48}$$

## B    EXAMPLE OF SUBOPTIMAL SOLUTIONS DUE TO INDEPENDENTLY OPTIMISED STACKED POOLING OPERATORS

Figure 6 shows two multilevel cluster assignments for the same graph. The clustering-based pooling methods are guided by a loss objective. For hierarchical clustering, this objective is stacked multiple times, here twice. Different from this common approach, we propose a hierarchical loss objective that considers the clusterings at both levels at the same time.

**(a)**                                                                              **(b)**

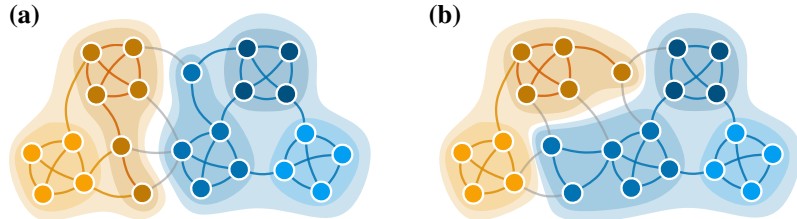

Figure 6: When stacked, all tested clustering methods agree that the clusters in (a) have lower loss and, thus, are better than those in (b). However, without stacking, the pooling operators consider the lower-level clusters in (b) superior to the lower-level clusters in (a). Because they do not consider the coarser clusters during the first step, they do not obtain the overall better assignment shown in (a). In contrast, our approach optimises the total assignment and returns (a) as the better solution, choosing a worse solution at the lower level for an overall better solution. The methods' loss values are presented in Table 4.

When considering the levels independently, the optimisation of the first level leads to a reduced solution space for the second level, such that the overall hierarchical solutions become worse. Table 4 presents the losses for both assignments shown in Figure 6. We note that, for all stacked methods, optimising the first level leads to solution (b). Considering the overall multilevel loss, which is the sum of the losses at the first and second level, suggests that solution (a) is in total better. Consequently, in the optimal solution, a good first-level assignment depends on the second-level assignment. This dependency is not modelled through the summation of loss terms. We propose MDL-Pool, a pooling operator that builds on the map equation and the minimum description length principle and considers the dependencies between different levels.

Table 4: Clustering loss for the different methods. Smaller is better. We compare the first-level clustering between both examples and the hierarchical clustering between both example.

| Pooling | First Level | | Two Levels Stacked | |
| Loss | $L_1(S_a)$ | $L_1(S_b)$ | $L_1(S_a) + L_2(S_a)$ | $L_1(S_b) + L_2(S_b)$ |
|---|---|---|---|---|
| BNPool (Rect Loss) | 230.715 | **226.715** | **179.547** | 183.547 |
| DiffPool | 0.015 | **0.014** | **1.145** | 1.180 |
| MinCut | -0.578 | **-0.600** | **-1.290** | -1.267 |
| JBPool | -0.996 | -0.996 | -1.991 | -1.991 |
| DMoN | -0.526 | **-0.547** | **-0.906** | -0.865 |
| MDL-Pool ($L^{(1)}$ stacked) | 3.632 | **3.548** | **5.478** | 5.605 |

## C    NOTES ON COMPLEXITY

Our approach follows the principles of the map equation and aims to detect clusters in graphs by identifying patterns in the graph's flow matrix (Blöcker et al., 2024). The flow matrix is calculated in a pre-processing step and, thus, does not extend the training duration. For undirected networks, it can be calculated in closed form; for directed networks, we calculate it with smart teleportation (Lambiotte & Rosvall, 2012; Blöcker et al., 2024) and a power iteration with a fixed number of iterations.

During the training, for all clustering-based methods, the most expensive operation is the pooling operation, $\mathbf{C}_\mathsf{M} = \mathbf{S}_\mathsf{M}^\top \mathbf{S}_\mathsf{m}^\top \mathbf{F} \mathbf{S}_\mathsf{m} \mathbf{S}_\mathsf{M}$. While we explicitly state this operation, it is also required for the other methods when they are stacked and, hence, applied multiple times. How long precisely the matrix multiplications to perform the pooling step take depends on the network's sparsity and the number of modules and submodules. When the graph is sparse, which is typically the case in empirical datasets, that is $|E| = \mathcal{O}(n)$, where $n = |V|$ is the number of nodes, and the number of clusters and superclusters, $m$ and $M$, respectively, are much smaller than the number of nodes, $m \ll n, M \ll n$, the complexity of the pooling operator is linear. The complexity becomes quadratic when the graph is dense $|E| = \mathcal{O}(n^2)$ or one of the cluster assignments contains nearly as many clusters as there are nodes. Even though our method strives for smaller assignments through Occam's razor, we guarantee a small number of clusters by fixing the maximum number to $c = 50$ such that $m < c$ and $M < c$. However, this setting can be adjusted as needed, for example, for extremely large graphs where more than 50 clusters are expected. By instead setting $c = \sqrt{n}$, we always obtain $\mathcal{O}(n)$ clusters, hence not increasing the runtime complexity. This value suits the empirical observations that many graphs have of the order of $\sqrt{n}$ many communities (Ghasemian et al., 2020).

# D    EVALUATION DETAILS

We conduct our experiments using the pooling evaluation framework provided by Castellana & Bianchi (2025), ensuring a standardised approach to analysing pooling methods. This allows us to reuse the baselines and datasets for consistency. All experiments are executed on an NVIDIA L40 GPU with 48GB VRAM.

For all experiments, we set the learning rate of Adam to $5 \times 10^{-4}$. Each experiment is repeated five times with different random seeds. For the classification tasks, the dataset is randomly split into 10% test data, with the remaining 90% further divided into 85% training and 15% validation data and set the maximum number of epochs to 1000. Early stopping is applied with a patience of 300 epochs, based on the classification metric on the validation set.

The clustering models consist of a single pre-layer followed by the pooling operator. The pre-layer is a one-layer GIN with an embedding size of 64. For the classification models, the architecture includes a two-layer GIN pre-layer, the pooling operator, a one-layer GIN post-layer, and an MLP for final predictions. Both the pre-layer and post-layer use an embedding size of 64. Dropout with a rate of 0.5 is applied, and the pooling ratio is set to 0.5 for all methods except BNPool and MDLPool, for which we fix $c = 50$. We consider a single pooling step for all baselines, due to the small graph sizes in the classification datasets and the observation that MDL-Pool rarely selects multiple pooling steps.

The code for reproducing the experiments is available at blinded. The implementations are provided as supplementary material to ensure anonymity during the double-blind review process and will be made publicly available after acceptance of the paper

## D.1    DATASETS

The datasets and most of the table content are obtained from Castellana & Bianchi (2025).

Table 5: Details of the used datasets.

| Dataset | #Samples | #Classes | Avg. #vertices | Avg. #edges | Vertex attr. | Vertex labels | Edge attr. |
|---|---|---|---|---|---|---|---|
| Citeseer | 1 | 6 (vertex) | 3,327.00 | 9,104.00 | 3,703 | yes | – |
| Community | 1 | 5 (vertex) | 400.00 | 5,904.00 | 2 | yes | – |
| Cora | 1 | 7 (vertex) | 2,708.00 | 10,556.00 | 1,433 | yes | – |
| DBLP | 1 | 4 (vertex) | 17,716.00 | 105,734.00 | 1,639 | yes | – |
| Pubmed | 1 | 3 (vertex) | 19,717.00 | 88,648.00 | 500 | yes | – |
| SBM | 1 | 5 (vertex) | 300.00 | 17,034.00 | 2 | yes | – |
| Collab | 5,000 | 3 (graph) | 74.49 | 4,914.43 | – | no | – |
| Colors3 | 10,500 | 11 (graph) | 61.31 | 91.03 | 4 | no | – |
| D&D | 1,178 | 2 (graph) | 284.32 | 1,431.32 | – | yes | – |
| Enzymes | 600 | 6 (graph) | 32.63 | 62.14 | 18 | yes | – |
| IMDB | 1,000 | 2 (graph) | 19.77 | 96.53 | – | no | – |
| molhiv | 41,127 | 2 (graph) | 25.5 | 27.5 | 9 | no | 3 |
| MUTAG | 188 | 2 (graph) | 17.93 | 19.79 | – | yes | – |
| Mutag. | 4,337 | 2 (graph) | 30.32 | 61.54 | – | yes | – |
| NCI1 | 4,110 | 2 (graph) | 29.87 | 64.60 | – | yes | – |
| Proteins | 1,113 | 2 (graph) | 39.06 | 72.82 | 1 | yes | – |
| RedditB | 2000 | 2 (graph) | 429.63 | 497.75 | – | no | – |

# E  ADDITIONAL RESULTS

Table 6: Community detection performance of soft-clustering -based pooling methods. (Top) We set the maximum number of clusters to match the ground truth, $c_{max} = |C|$. (Bottom) We consider the number of clusters unknown, setting $c_{max} = 50$. We list the average ONMI over 5 runs. A node is assigned to one or multiple communities if the value in the assignment matrix is at least $1/c_{max}$. Smaller parts are discarded as noise. The median number of found communities, $\tilde{c}$, is shown in parentheses. Overall best results are red.

| | Method | CiteSeer $|C| = 6$ | Community $|C| = 5$ | Cora $|C| = 7$ | DBLP $|C| = 4$ | PubMed $|C| = 3$ | SBM $|C| = 5$ |
|---|---|---|---|---|---|---|---|
| $c_{max} = |C|$ | BNPool | $0.9 \pm 0.1$ (6) | $25.1 \pm 8.0$ (5) | $2.4 \pm 1.5$ (6) | $11.5 \pm 1.1$ (4) | $7.5 \pm 0.2$ (3) | $55.8 \pm 1.5$ (3) |
| | DiffPool | $7.3 \pm 1.2$ (6) | $64.9 \pm 3.0$ (5) | $16.0 \pm 5.3$ (7) | $6.6 \pm 0.4$ (4) | $6.7 \pm 1.0$ (3) | $\mathbf{100.0} \pm 0.0$ (5) |
| | DMoN | $\mathbf{8.7} \pm 6.2$ (6) | $79.4 \pm 10.3$ (5) | $13.1 \pm 7.9$ (7) | $9.2 \pm 4.9$ (4) | $11.5 \pm 6.9$ (3) | $\mathbf{100.0} \pm 0.0$ (5) |
| | JBGNN | $5.6 \pm 6.6$ (6) | $\mathbf{90.6} \pm 2.3$ (5) | $6.7 \pm 4.8$ (7) | $10.0 \pm 3.8$ (4) | $2.9 \pm 3.5$ (3) | $92.5 \pm 10.3$ (5) |
| | MinCut | $7.7 \pm 3.9$ (6) | $87.0 \pm 1.0$ (5) | $\mathbf{20.7} \pm 4.0$ (7) | $\mathbf{16.5} \pm 1.2$ (4) | $9.3 \pm 3.4$ (3) | $\mathbf{100.0} \pm 0.0$ (5) |
| | MDL-Pool | $4.4 \pm 3.2$ (6) | $71.2 \pm 13.3$ (4) | $19.7 \pm 5.1$ (6) | $10.2 \pm 5.6$ (4) | $\mathbf{15.6} \pm 6.0$ (3) | $92.6 \pm 10.1$ (5) |
| $c_{max} = 50$ | BNPool | $0.6 \pm 0.1$ (42) | $12.4 \pm 1.8$ (28) | $1.6 \pm 1.0$ (40) | $4.3 \pm 0.8$ (36) | $4.5 \pm 1.8$ (31) | $14.6 \pm 0.8$ (24) |
| | DiffPool | $2.2 \pm 0.6$ (50) | $49.7 \pm 1.7$ (38) | $7.1 \pm 0.3$ (50) | $3.8 \pm 0.2$ (50) | $2.4 \pm 0.3$ (50) | $89.4 \pm 3.2$ (9) |
| | DMoN | $0.0 \pm 0.0$ (50) | $17.2 \pm 2.1$ (50) | $0.0 \pm 0.0$ (50) | $1.6 \pm 0.1$ (50) | $1.2 \pm 0.5$ (50) | $92.6 \pm 1.4$ (50) |
| | JBGNN | $0.8 \pm 1.3$ (45) | $28.6 \pm 1.0$ (27) | $6.1 \pm 1.7$ (46) | $4.8 \pm 2.1$ (49) | $0.4 \pm 0.4$ (49) | $88.2 \pm 6.3$ (12) |
| | MinCut | $\mathbf{3.2} \pm 1.5$ (50) | $4.4 \pm 4.4$ (24) | $5.4 \pm 0.7$ (50) | $1.5 \pm 0.4$ (36) | $1.1 \pm 0.6$ (35) | $95.4 \pm 1.2$ (38) |
| | MDL-Pool | $2.0 \pm 1.3$ (41) | $\mathbf{62.1} \pm 8.8$ (17) | $\mathbf{14.5} \pm 3.8$ (28) | $\mathbf{5.0} \pm 2.2$ (38) | $\mathbf{8.9} \pm 1.5$ (45) | $\mathbf{100.0} \pm 0.0$ (5) |

# F  Implementation Details and Ablation Study

This section provides implementation details and design choices essential for reproducing the results. We validate these choices through an ablation study presented in Table 7.

**Pooling Depth**   We limit our method to at most two pooling operations. While the approach can be extended to more layers, we observed that the graphs in the classification datasets are small enough that two layers suffice. This observation is supported by the measures in Figure 3 and the empirical argument that real-world data often contains approximately $\sqrt{n}$ communities (Ghasemian et al., 2020), where $n$ is the number of nodes in the graph, leading to a negligible number of clusters in deeper levels. In the ablation study, we evaluate the impact of fixing the number of levels to a specific depth (1-LVL or 2-LVL) instead of using adaptive model selection. Furthermore, we extend the adaptive pooling depth up to three levels (Base+3-LVL) to verify that 3 levels get not selected due to the negligible community size even though the model would be capable of selecting it.

**Detached Backpropagation**   During backpropagation, all pooling operators are trained. However, the backward path is detached after the pooling operators to prevent them from influencing the pre-layer that generates the embeddings $\mathbf{H}$. This ensures that unselected pooling operations do not affect the pre-layer. The ablation study demonstrates the impact of this design choice.

**Soft Map Equation**   The original map equation was designed for hard cluster assignments, where each node belongs to exactly one cluster. With the assignment matrix, nodes can be partially assigned to multiple clusters. This introduces two options for adapting the map equation-based loss (Blöcker et al., 2024): (1) Distribute the node's contribution to the description length proportionally across the containing modules. (2) Weight the node's contribution such that each module incurs the full impact of the node, making soft assignments more expensive. In the main work, we adopt the first option, as the second option creates a loss landscape where transitioning between assignments via intermediate soft states becomes prohibitively expensive. The ablation study evaluates the second option for comparison.

**Fallback for Single-Clusters**   For model selection, our approach compares the description lengths of different pooling depths and prefers the shallower operator when two or more description lengths are the same. However, due to numerical imprecision in soft-assignments, the deeper operator might occasionally have a slightly lower description length when all top-level nodes are assigned to a single top-level community, effectively replicating the shallower operator. In this case, we explicitly fall back to the shallower operator. In the ablation study, we test whether not falling back in such cases affects the performance.

**Assignment Network Architecture**   We use an MLP to generate cluster assignments from the embeddings. However, a GNN, such as a GIN, is another viable option. In the ablation study, we replace the MLP with a GIN to evaluate its impact on performance.

Table 7: Results of the ablation study with the best results for each dataset marked in bold.

| Pooler | COLLAB | COLORS-3 | DD | ENZYMES | IMDB-B | MUTAG | Mutag. | NCI1 | PROTEINS | REDDIT-B | molhiv (auroc) |
|---|---|---|---|---|---|---|---|---|---|---|---|
| Base | 76.3 ± 0.9 | 87.2 ± 1.8 | **79.7** ± 2.5 | 39.3 ± 3.2 | **77.2** ± 5.4 | 85.7 ± 8.7 | 80.0 ± 2.0 | 79.0 ± 1.2 | **76.1** ± 5.5 | 91.6 ± 1.1 | 75.2 ± 2.0 |
| 1-LVL | 68.9 ± 6.0 | 86.5 ± 1.2 | 77.3 ± 2.0 | **41.3** ± 5.2 | 76.0 ± 5.1 | **90.0** ± 8.1 | 80.5 ± 0.8 | 78.0 ± 1.7 | 75.9 ± 4.6 | 91.3 ± 1.8 | **76.3** ± 1.0 |
| 2-LVL | 66.7 ± 6.6 | 87.3 ± 1.3 | 73.7 ± 2.2 | 31.0 ± 3.0 | 71.6 ± 2.6 | 88.6 ± 8.1 | 79.4 ± 1.5 | 77.6 ± 1.3 | 74.3 ± 0.9 | 90.4 ± 1.9 | 65.9 ± 1.4 |
| GIN | 75.2 ± 1.3 | 88.9 ± 1.2 | 76.1 ± 5.1 | 36.0 ± 3.7 | 73.6 ± 7.1 | **90.0** ± 3.9 | **81.0** ± 1.4 | **79.4** ± 1.5 | 74.7 ± 4.8 | 91.1 ± 1.5 | 76.1 ± 1.4 |
| Attached BP | **77.0** ± 1.2 | 87.5 ± 1.9 | 73.2 ± 4.2 | 36.7 ± 6.7 | 75.6 ± 8.6 | 85.7 ± 5.1 | 80.1 ± 2.6 | 77.7 ± 1.5 | 75.3 ± 6.5 | **91.9** ± 1.1 | 74.8 ± 1.4 |
| Soft Loss | 76.2 ± 0.8 | **89.0** ± 2.1 | 75.4 ± 4.1 | 40.3 ± 8.4 | 74.4 ± 4.8 | 85.7 ± 5.1 | 79.4 ± 2.7 | 78.6 ± 2.4 | 75.5 ± 5.2 | 90.9 ± 1.0 | 75.6 ± 1.9 |
| Base + 3-LVL | 70.6 ± 3.2 | 76.4 ± 8.2 | 74.3 ± 4.3 | 38.6 ± 11.3 | 71.6 ± 5.3 | 88.1 ± 3.7 | 80.1 ± 1.7 | 79.0 ± 2.1 | 72.9 ± 3.2 | 89.9 ± 1.9 | 75.6 ± 2.6 |

Overall, we find that the different variants of MDL-Pool can produce better results than our base model in some cases. However, the base model's performance is almost always within the standard deviation of the variant's performance.

