# OpenReview forum: "MDL-Pool: Adaptive Multilevel Graph Pooling Based on Minimum Description Length"
_ICLR.cc/2026/Conference — Submitted to ICLR 2026_

### Official Review · Reviewer_5Qqa · 2025-10-28

**Soundness:** 3
**Presentation:** 2
**Contribution:** 2
**Rating:** 4
**Confidence:** 3

**Summary:**

This paper introduces MDL-Pool, a differentiable graph pooling method that automatically selects the optimal pooling depth for each graph by minimizing the description length—computed via the map equation—of hierarchical cluster assignments.
The method adaptively compresses graph structures and demonstrates effectiveness on both community detection and graph classification tasks.

**Strengths:**

- Explicit modeling of interdependencies across different hierarchical levels, as opposed to stacking independent pooling layers.

**Weaknesses:**

- Limited performance improvement
The authors note that MDL-Pool achieves state-of-the-art results only “in two of the eleven scenarios” and explicitly acknowledge that “we do not find a clear winner for graph classification.” This suggests that the overall empirical gain is modest.
- Lack of dataset-specific performance analysis
The paper does not explore which dataset characteristics (e.g., average graph size) affect the method’s relative performance.
A correlation study between these properties and MDL-Pool’s performance would strengthen the empirical section.
- Unclear practical advantage of adaptive depth
Adaptive depth is a central contribution, yet the paper reports that “the maximum chosen depth was two” and that “graphs in the classification datasets are small enough that two layers suffice”.
For single-graph tasks (community detection) or datasets where most graphs use depth 0 or 1, the advantage over fixed-depth configurations remains unclear, especially given that the model performance is comparative to other fixed-depth baselines.

- Computational cost and parameter-free claim
The paper emphasizes that MDL-Pool is “parameter-free,” yet it is also “limited to at most two pooling operations”
The actual computational savings compared to conventional hyperparameter tuning are therefore uncertain, given that MDL
Reporting runtime or complexity comparisons with fixed-depth GNN baselines would make the claim more convincing.

**Questions:**

- Could you provide a quantitative analysis correlating dataset characteristics (e.g., average node/edge count, modularity) with the relative performance of MDL-Pool?
- Do the graph-pooling results align with domain knowledge? For example, in the D&D dataset, do amino acids belonging to the same secondary structure fall within the same cluster?

---

> ### Author Response · Authors · 2025-11-21
>
> Thank you for your constructive feedback and the positive words on our work.
>
> > Could you provide a quantitative analysis correlating dataset characteristics (e.g., average node/edge count, modularity) with the relative performance of MDL-Pool?
>
> Of course, we would be happy to add such a table. We have analysed the correlation of MDL-Pool's relative performance (relative to the best and average baseline performance, respectively) with the average number of nodes, average number of edges, and the clustering coefficient using Spearman's rank correlation coefficient. We found that the correlation between the average number of edges per dataset and the relative performance improvement of MDL-Pool is significant, suggesting that MDL-Pool tends to perform better in graphs with more edges.
>
> First, the data:
> ||COLLAB|COLORS-3|D&D|ENZYMES|IMDB-B|MUTAG|Mutag.|NCI1|PROTEINS|REDDIT-B|molgiv|
> |-|-|-|-|-|-|-|-|-|-|-|-|
> |MDL-Pool|76.3|87.2|79.7|39.3|77.2|85.7|80.0|79.0|76.1|91.6|75.2|
> |best baseline|77.1|97.1|78.3|45.0|76.8|90.0|82.3|80.6|77.1|93.0|77.4|
> |mean baseline|74.09|87.42|73.92|39.04|74.88|87.26|80.59|78.59|75.74|90.59|75.87|
> |avg #nodes|74.49|61.31|284.32|32.63|19.77|17.93|30.32|29.87|39.06|429.63|25.51|
> |avg #edges|4914.43|182.05|1431.32|124.27|193.06|39.59|61.54|64.6|145.63|995.51|54.94|
> |avg clustering|18.82|15.15|71.07|8.14|5.17|4.23|7.33|7.22|9.77|107.18|6.13|
>
> And the results: Spearman correlation coefficient r and p-values.
> | |avg. #nodes|avg. #edges|avg clustering|
> |-|-|-|-|
> | rel. improvement against best baseline    |r=0.31 (p=0.36)|**r=0.66 (p=0.03)**|r=0.31 (p=0.1)|
> | rel. improvement against average baseline |r=0.53 (p=0.1)|**r=0.87 (p<0.0005)**|r=0.52 (p=0.1)|
>
>
> > Do the graph-pooling results align with domain knowledge? For example, in the D&D dataset, do amino acids belonging to the same secondary structure fall within the same cluster?
>
> To be honest, we cannot say because we are not domain experts in those fields. However, we appreciate the reviewer's suggestion to investigate this and get input from someone who is a domain expert. However, upon visual inspection, the results seem sensible (Figure 4).

---

> > ### Comment · Reviewer_5Qqa · 2025-11-24
> >
> > Thank you for the detailed response.
> >
> > The correlation analysis you added (particularly the strong relationship with average edge count) is helpful and meaningfully clarifies how MDL-Pool behaves across datasets with different structural characteristics.
> >
> > Regarding the D&D domain-specific question, I understand the limitation.

---

### Official Review · Reviewer_iTTL · 2025-10-29

**Soundness:** 3
**Presentation:** 2
**Contribution:** 2
**Rating:** 4
**Confidence:** 3

**Summary:**

The paper introduces MDL-Pool, a new adaptive, multilevel graph pooling operator grounded in information theory. It applies the map equation to jointly optimize cluster assignments across all hierarchical levels, explicitly modeling interdependencies between them. Unlike existing hierarchical pooling methods that fix the number of pooling layers, MDL-Pool dynamically selects the optimal pooling depth per graph using the MDL principle.

**Strengths:**

- The integration of the MDL principle and map equation into deep graph pooling is well-motivated. It provides a principled way to address overfitting and model complexity while enhancing interpretability.
- The proposed multilevel loss seamlessly integrates hierarchical information, overcoming optimization issues caused by layer-wise independence in stacked pooling.
- The MDL framework naturally implements Occam’s razor, removing the need for hyperparameter tuning for cluster count or levels.

**Weaknesses:**

- The MDL-based loss focuses on topological structure and does not fully leverage node features in evaluating community quality, which might reduce performance on feature-dominant tasks.
- Experiments show most graphs select only one or two pooling levels; it remains unclear whether MDL-Pool is beneficial in tasks with truly deep hierarchies.
- The computation of multilevel flow matrices has quadratic cost in graph size, which may hinder scalability to very large graphs. No experiments on large-scale datasets are shown.

**Questions:**

See the above weaknesses.

---

> ### Author Response · Authors · 2025-11-21
>
> We would like the reviewer for the positive words and for highlighting the strengths of our work.
>
> > The MDL-based loss focuses on topological structure and does not fully leverage node features in evaluating community quality, which might reduce performance on feature-dominant tasks.
>
> You are right, the map equation, and, consequently, our adapted multilevel loss formulation for graph pooling do not directly consider node features. This is because the map equation is a topological measure for community detection. Instead, node features enter the optimisation via the supervised training for the downstream graph classification task, thus, contributing to identifying clusters. However, there are generalisations of the map equation that consider node features/metadata [1,2], and it could be a promising research direction to include node features directly into the unsupervised loss function---thank you for this suggestion.
>
> > Experiments show most graphs select only one or two pooling levels; it remains unclear whether MDL-Pool is beneficial in tasks with truly deep hierarchies.
>
> Indeed, and we believe that this is due to the characteristics of the datasets. It seems as though the graphs included in the datasets do not exhibit hierarchical structure at a greater depth than 2 in most cases, which is still more than previous methods have considered. In our ablation study (Appendix F), we have also explored a possible depth of up to 3, but we did not see an improvement for the tested datasets. And, unfortunately, we are not aware of synthetic datasets with deeper hierarchies. Creating such datasets may also be an interesting research direction. Otherwise, we expect that datasets with graphs that have a hierarchical structure will arise over the coming years.
>
> > The computation of multilevel flow matrices has quadratic cost in graph size, which may hinder scalability to very large graphs. No experiments on large-scale datasets are shown.
>
> Actually, computing the flow matrices is essentially the same as computing PageRank, and that can be done efficiently with a power iteration, scaling to networks that have the size of the internet. With a power iteration using a sparse matrix representations, the complexity is $\mathcal{O}(kmn)$, where $n$ is the number of nodes, $m$ the number of edges, and $k$ the number of iterations. Moreover, computing the flow matrix $\mathbf{F}$ can be seen as a preprocessing step, which only needs to be done once per dataset.
>
> [1] Emmons, Scott, and Peter J. Mucha. "Map equation with metadata: Varying the role of attributes in community detection." Physical Review E 100.2 (2019): 022301.\
> [2] Bassolas, Aleix, et al. "Mapping nonlocal relationships between metadata and network structure with metadata-dependent encoding of random walks." Science Advances 8.43 (2022): eabn7558.

---

### Official Review · Reviewer_GEGj · 2025-11-03

**Soundness:** 3
**Presentation:** 2
**Contribution:** 3
**Rating:** 6
**Confidence:** 4

**Summary:**

This paper proposes a minimum-description-based graph pooling model to learn strong graph representations for different-sized networks. With the mapping function, the model encodes the vertices' interdependencies. in different hierarchical levels that help cluster the network according to its volume and make graph learning effective.

**Strengths:**

The map equation is beneficial for observing the overall networks and for relevant clustering.
The clustering helps for balanced training to fit the model in downstream graph analytics tasks.
MDL is beneficial because it detects the depth of the input graph, which assists in effective hierarchical graph learning.
The comprehensive result is better than other baselines

**Weaknesses:**

In the experiment, the authors did not mention the hyperparameter's impact on the model.
The manuscript does not provide runtime details. Is minimum description length feasible on large volume datasets?
The optimization of map equations involves nested matrix operations, which can result in a computationally heavy model. Please check the model's runtime with respect to simpler pooling operations like Top-kPool and SAGPool.
In the case of community detection, the datasets are very sparse. Is the model suitable to cluster denser graphs (like Amazon Photo and Physics)? In this case, how does it perform over the other baselines?

**Questions:**

Is minimum description length feasible on large volume datasets?  Please observe the community detection on Amazon Photos, Physics.
How much more efficient is the model compared to simpler pooling operations like Top-kPool, SAGPool, GMT, etc.? Does MDLPool outperform these models?
How does the technique provide expressivity in the model ? could you please show some formal reasoning or visualization?

---

> ### Author Response · Authors · 2025-11-21
>
> Thank you for the positive evaluation of our work and highlighting that our results are "better than other baselines".
>
> > Is minimum description length feasible on large volume datasets?
>
> Indeed, approaches based on the minimum description length principle are applicable to large-scale datasets. On a high level, our work can be seen as combining an information-theoretic loss function with graph neural networks, which is similar to what previous works, such as DMoN or DiffPool have done. Please also refer to Appendix C, where we discuss the complexity of our approach. The gist of it is that, in practice, graphs are typically sparse and contain much fewer modules $m$ than nodes $n$, i.e., $m \ll n$, and also much fewer links than theoretically possible, i.e., $|E| \ll n^2$, which leads to linear runtime (in the number of edges). In the worst case, when the graph is dense and there are many edges, i.e., $|E| = \mathcal{O}(n^2)$, the complexity becomes quadratic. However, the same holds for other methods. As can be seen in Appendix D, we have tested MDL-Pool on graphs with up to approx. 20,000 nodes or 105,000 edges.
>
> > How much more efficient is the model compared to simpler pooling operations like Top-kPool, SAGPool, GMT, etc.?
>
> As mentioned above, the complexity of MDL-Pool is similar to that of other clustering-based methods, such as DMoN. However, score-based methods, such as Top-k, are more efficient because they essentially only "filter" the graph to find important nodes, but without coarse-graining the graph to obtain a higher-level representation. In case our paper is accepted, we will add a table with recorded runtimes for the different methods to Appendix C.
>
> > How does the technique provide expressivity in the model?
>
> We are a bit unsure what the reviewer means exactly. Could you elaborate? We are not sure if you mean this, but our method does not affect the expressivity of the individual GNN layers in the Weisfeiler-Leman sense. However, clustering-based methods enable longer-range communication because they aggregate the nodes within a community and construct a coarse-grained version of the graph on which they perform message passing. Previous results show that, while score-based methods, such as Top-k and SAGPool, reduce the expressivity of GNNs, clustering-based methods generally do not affect them [1].
>
> [1] Bianchi, Filippo Maria, and Veronica Lachi. "The expressive power of pooling in graph neural networks." Advances in neural information processing systems 36 (2023): 71603-71618.

---

### Official Review · Reviewer_8QYD · 2025-11-03

**Soundness:** 2
**Presentation:** 3
**Contribution:** 2
**Rating:** 2
**Confidence:** 4

**Summary:**

This paper proposes MDL-Pool, an adaptive multilevel graph pooling operator based on the Minimum Description Length (MDL) principle. The method integrates the map equation into graph neural networks (GNNs) to model hierarchical dependencies between pooling levels. Unlike conventional stacked pooling approaches that fix depth and ignore inter-level dependencies, MDL-Pool formulates a joint loss to optimize clusters across all hierarchical levels and automatically selects the optimal depth per graph. The authors evaluate MDL-Pool on both community detection and graph classification benchmarks.

**Strengths:**

1. The method automatically determines the optimal pooling depth per graph instance, addressing a long-standing hyperparameter issue in hierarchical pooling.
2. The paper provides experiments on both synthetic and real-world datasets, including ablations on architecture variants and pooling depths.

**Weaknesses:**

Limited performance gain: In Tables 2 and 3, MDL-Pool does not consistently outperform baselines. For community detection, results are comparable or even worse than baselines on several datasets. Similarly, in graph classification, MDL-Pool’s average accuracy is not higher than several baselines, indicating limited empirical advantage.

Insufficient justification of benefits: While the motivation is sound, the claimed benefits (interdependency modeling and adaptive depth) are not strongly supported by quantitative evidence. The paper should include ablation or visualization explicitly demonstrating that modeling interdependencies leads to measurable improvement.

Unclear parameter selection: Section 4.1 mentions “up to l levels,” but the criterion for selecting the number of levels is not clearly described. How the model avoids overfitting or underfitting different depths needs more elaboration.

Choice of cmax = 50: In Table 2, the authors fix the number of clusters to 50, which seems far from the ground-truth number of communities (e.g., 3–7). This may bias the results. The paper should report results when cmax is closer to the true number (e.g., 10) to assess robustness.

Lack of comparative summary metrics: For Table 3, it would be informative to include an overall metric, such as the average rank or mean relative improvement across datasets, to better illustrate general trends rather than per-dataset fluctuations.

Questionable realization of motivation: The introduction claims that previous works ignore interdependencies between hierarchical structures, but it remains unclear whether MDL-Pool effectively learns such interdependencies rather than simply aggregating multi-level losses. Experimental evidence (e.g., hierarchical attention visualization or gradient correlation analysis) is lacking.

**Questions:**

see above

---

> ### Author Response · Authors · 2025-11-21
>
> Thank you for the constructive feedback and highlighting the strengths of our work.
>
> > In Tables 2 and 3, MDL-Pool does not consistently outperform baselines.
>
> You're right, MDL-Pool doesn't outperform all baselines in all datasets. However, the same is true for the baselines: None of them outperforms all other methods in all datasets. Nevertheless, MDL-Pool outperforms the baselines on some of the datasets, and performs on par with them on the remaining datasets, which is no small feat.
>
> > While the motivation is sound, the claimed benefits (interdependency modeling and adaptive depth) are not strongly supported by quantitative evidence.
>
> We respectfully disagree. We show in Figure 3 that MDL-Pool chooses the optimal pooling depth, where we set the maximum possible depth to 2. Moreover, we provide ablation studies in Appendix F, where, amongst other experiments, we
>
> 1. fix the pooling depth to 1
> 2. fix the pooling depth to 2
> 3. use a GIN as the base model
> 4. allow a pooling depth of up to 3
>
> Moreover, Appendix B includes a concrete example where we show how the solutions differ between (i) merely stacking pooling operators and (ii) considering interdependencies.
>
> > Section 4.1 mentions “up to l levels,” but the criterion for selecting the number of levels is not clearly described. How the model avoids overfitting or underfitting different depths needs more elaboration.
>
> MDL-Pool avoids over- and underfitting via the minimum description length (MDL) principle, encoded in the multilevel clustering loss we derived from the map equation. Based on the MDL principle, the map equation makes a tradeoff between (i) many small and (ii) few large modules; this principle is applied over the multiple levels in the hierarchy---importantly, in an unsupervised fashion. Moreover, Eq. (6) states clearly how the optimal depth is selected: we learn pooling operators for all depths up to the maximum depth separately, and choose, via the MDL principle, the optimal depth for each graph instance. Again, in an unsupervised fashion.
>
> > In Table 2, the authors fix the number of clusters to 50, which seems far from the ground-truth number of communities (e.g., 3–7). This may bias the results. The paper should report results when cmax is closer to the true number (e.g., 10) to assess robustness.
>
> Perhaps there is a misunderstanding. In practice, we don't know the "true" number of clusters, and wish to rely on the clustering method to determine this property. To simulate that, we chose a max number of clusters well above the "true" number. However, we agree that there are valid reasons for choosing a more restrictive number of clusters, e.g., for computational efficiency. Following empirical results that the number of clusters scales as $\sqrt{n}$, where $n$ is the number of nodes [1], $c_\text{max} = 50$ would suggest about $2500$ nodes; consequently, setting $c_\text{max} = 50$ can be seen as a better estimate of the "true" number of clusters than $c_\text{max} = 10$ for graphs with more than $2500$ nodes.
>
> But: if we set $c_\text{max}$ to the "true" number of clusters, we provide the model with information typically not available in practice, preventing us from testing whether it can determine the correct number of clusters.
> In any case, as shown in Table 2, we found that most methods return much more than the correct number of clusters for $c_\text{max} = 50$. Please also note that Table 2 *also* contains results for setting $c_\text{max}$ to the "true" number of clusters.
>
> > For Table 3, it would be informative to include an overall metric, such as the average rank or mean relative improvement across datasets, to better illustrate general trends rather than per-dataset fluctuations.
>
> Thank you for the suggestion, this would be a useful addition for better interpretability of our results.
>
> > The introduction claims that previous works ignore interdependencies between hierarchical structures, but it remains unclear whether MDL-Pool effectively learns such interdependencies [...]. Experimental evidence [...] is lacking.
>
> Consider a random walker on the graph. If the random walker remains within the same community, we use one codeword to describe the step. If the nodes are in different communities, we use at least three codewords: one each to exit the current community, enter the new community, and visit the target node. With multiple levels in the hierarchy, we may need to "exit" up to the top level first, and then descend several levels to reach the target community. This behaviour is encoded in the map equation, describing the interdependent coding structures across the different levels. In case our paper is accepted, we will extend the descriptions of the equations in Appendix A to point this out more explicitly.
>
> [1] Ghasemian, Amir, Homa Hosseinmardi, and Aaron Clauset. "Evaluating overfit and underfit in models of network community structure." IEEE Transactions on Knowledge and Data Engineering 32.9 (2019): 1722-1735.

---

> > ### Comment · Reviewer_8QYD · 2025-11-27
> > **re rebuttal**
> >
> > Thank you for the detailed responses. Several of my earlier concerns have been addressed, and I appreciate the clarifications provided. I will increase my score to 4.
> >
> > However, I still find it unclear how the proposed method effectively learns and utilises interdependencies between hierarchical structures in practice. In addition, as other reviewers have noted, the scalability of MDL-Pool to large graphs remains insufficiently demonstrated

---

### Meta-Review · Area_Chair_yZaz · 2026-01-07

**Summary:**

The reviewers raised concerns about the limited performance gain,  unclear experimental setups, insufficient ablation studies, and limited scalability. The main concerns in the experimental evaluation remain unresolved after the rebuttal. Therefore, I recommend rejecting this paper.

**Reviewer Concerns:**

- Reviewer 8QYD: Limited performance gain, Unclear experimental setups, insufficient evaluation metrics, limited motivation
- Reviewer GEGj: Insufficient ablation studies, lack of runtime details, inappropriate choices of datasets
- Reviewer iTTL: Insufficient experiments, limited scalability to very large graphs
- Reviewer 5Qqa: Limited performance improvement, Lack of dataset-specific performance analysis

**Reviewer Scores:**

- Reviewer 8QYD: No
- Reviewer GEGj: No
- Reviewer iTTL: No
- Reviewer 5Qqa: No

---

### Decision · Program_Chairs · 2026-01-26

Reject